# The retrieval of snow properties from SLSTR/ Sentinel-3 - part 1: method description and sensitivity study

**Linlu Mei[1], Vladimir Rozanov[1], Christine Pohl[1], Marco Vountas[1]**, **John P. Burrows[1]**

[1] Institute of Environmental Physics, University of Bremen, Germany

**Abstract**

The eXtensible Bremen Aerosol/cloud and surfacE parameters Retrieval (XBAER) algorithm has been designed for the Top-Of-Atmosphere reflectance measured by the Sea and Land Surface Temperature Radiometer (SLSTR) instrument onboard Sentinel-3 to derive snow properties: Snow Grain Size (SGS), Snow Particle Shape (SPS) and Specific Surface Area (SSA) under cloud-free conditions. This is the first part of the paper, to describe the retrieval method and the sensitivity study. Nine pre-defined SPSs (aggregate of 8 columns, Drontal, hollow bullet rosettes, hollow column, plate, aggregate of 5 plates, aggregate of 10 plates, solid bullet rosettes, column) are used to describe the snow optical properties. The optimal SGS and SPS are estimated iteratively utilizing a Look-Up-Table (LUT) approach. The SSA is then calculated using another pre-calculated LUT for the retrieved SGS and SPS. The optical properties (e.g., phase function) of the ice crystals can reproduce the wavelength-dependent/angular-dependent snow reflectance features, compared to laboratory measurements. A comprehensive study to understand the impact of aerosol, SPS, ice crystal surface roughness, cloud contamination, instrument spectral response function, snow habit mixture model, and snow vertical inhomogeneity on the retrieval accuracy of snow properties has been performed based on SCIATRAN radiative transfer simulations. The main findings are (1) Snow angular and spectral reflectance feature can be described by the predefined ice crystal properties only when both SGS and SPS can be optimally and iteratively obtained; (2) The impact of ice crystal surface roughness plays minor effects on the retrieval results; (3) SGS and SSA show an inverse linear relationship; (4) The retrieval of SSA assuming non-convex particle shape, compared to

convex particle (e.g. sphere) shows larger results; (5) Aerosol/cloud contamination due to unperfected atmospheric correction and cloud screening introduces underestimation of SGS, "inaccurate" SPS and overestimation of SSA; (6) The impact of instrument spectral response function introduces an overestimation on retrieved SGS, an underestimation on retrieved SSA and no impact on retrieved SPS; (7) The investigation, by taking a ice crystal particle size distribution and habit mixture into account, reveals that XBAER retrieved SGS agrees better with the mean size, rather than the mode size for a given particle size distribution.

## 1 Introduction

Snow properties such as snow albedo, Snow Grain Size (SGS), Snow Particle Shape (SPS), Specific Surface Area (SSA), snow purity (Warren and Wiscombe, 1980; Painter et al., 2003; Hansen and Nazarenko, 2004; Taillandier et al., 2007; Gallet et al., 2009; Battaglia et al., 2010; Gardner et al., 2010; Domine et al., 2011; Liu et al., 2012; Qu et al., 2015; Baker et al., 2019; Pohl et al., 2020a) show large variabilities temporally and spatially (Kukla et al., 1986). They play important roles in the global radiation budget, which is critical to some well-known phenomenon such as the Arctic amplification (Serreze and Francis, 2006; Domine et al., 2019). Satellites offer an effective way to understand the surface-atmosphere processes and corresponding feedback mechanisms on the regional, continental and/or global scales (Konig et al., 2001; Pope et al., 2014). Satellite derived snow products (e.g., SGS, SPS, and SSA) are particularly important for short-term hydrological, meteorological and climatological modelling (Livneh et al., 2009). A high-quanlity snow property data product can also be applied to derive Aerosol Optical Thickness (AOT) over cryosphere (Mei et al., 2020a). High-quality satellite derived snow products and their by-products are also important for the creation of long-term "Climate Data Records" (SSMC, 2014), which enable a better investigation and interpretation concerning global climate change (Konig et al., 2001). However, both the definition and the corresponding data accuracy of SGS are poor (Langlois et al., 2020) while there is no existing SPS satellite product. The lack of good information on SGS and SPS leads to low quality of SSA (Gallet et al., 2009). The accuracy of SGS, SPS and SSA limits the model

performance for the prediction of snow properties related to climate change issues. Lack of
information of SGS and SPS also restricts the accuracy of snow bidirectional reflectance
estimation, which further limits the retrieval possibilities of aerosol and cloud properties above
snow (Mei et al., 2020a, 2020b).
A comprehensive overview of remote sensing of SGS, SPS, and SSA can be found in
many previous publications (e.g., Li et al., 2001; Stamnes et al., 2007; Koren, 2009; Lyapustin
et al., 2009; Dietz et al., 2012; Wiebe et al., 2013; Frei et al., 2012; Mary et al., 2013;
Kokhanovsky, et al., 2019; Xiong et al., 2018). The variation of SGS leads to the large
variability of Top Of Atmosphere (TOA) reflectance in NIR/SWIR spectral ranges while SPS
shows a strong impact on TOA reflectance at visible channels (Warren and Wiscombe, 1980).
Different retrieval algorithms have been developed for different instruments. For instance, the
MODIS Snow Covered-Area and Grain size (MODSCAG) retrieval algorithm and Multi-Angle
Implementation of Atmospheric Correction (MAIAC) algorithm have been used to derive SGS
using MODIS and VIIRS instruments (Painter et al., 2003; 2009; Lyapustin et al., 2009).
Snow particle shape is another important parameter which affects the estimation of snow
properties, such as albedo (Räisänen et al., 2017; Flanner and Zender, 2006), because ice
crystals with different shapes have different optical properties (Jin et al., 2008; Yang et al.,
2013). The absorption and extinction cross-sections of an ice crystal can be described as a
function of size, shape, and refractive index at a given wavelength (van de Hulst 1981;
Mischenko et al., 2002 and references therein). Natural snow consists of grains, depending on
temperature, humidity, and meteorological conditions, which have numerous different shapes
(Nakaya, 1954). SPSs have been classified into different categories, the classification has been
increased from 21 (Nakaya and Sekido, 1938) to 121 categories (Kikuchi et al., 2013). Although
spherical shape assumption is typically used for field measurements (Flanner and Zender, 2006;
Donahue et al., 2020), this approximation is not recommended to be used in retrieval algorithms
of satellite measurements because it leads to large differences between observed and simulated
wavelength-dependent snow bidirectional reflectance, especially at visible wavelengths
(Leroux and Fily et al., 1998; Aoki et al., 2000; Jin et al., 2008; Dumont et al., 2010; Libois et
al., 2013). Improper wavelength-dependent snow bidirectional reflectance caused by a
predefined SPS leads to low-quality satellite retrieval results. Some attempts to derive SPS in
the ice cloud can be found in previous publications (McFarlane et al., 2005; Cole et al., 2014).
According to Legagneux et al., (2002), SSA is defined as the surface area of ice crsytal
per unit mass, i.e., SSA = $A_t/\rho V$, where $A_t$ and V are total surface area and volume, respectively,
$\rho$ is the ice density. SSA includes information on both SGS and SPS and it is often used to
describe the surface area available for chemical processes (Taillandier et al., 2007; Domine et
al., 2011; Yamaguchi et al., 2019). SSA is reported to have a good relationship with snow
spectral albedo at the short wave infrared wavelengths (Domine et al., 2007). Optical methods
are routinely used to measure SSA in the field (Gallet et al., 2009). Empirical equations have
been proposed to describe the change of SSA (Legagneux and Domine, 2005; Taillandier et al.,
2007). Few attempts have been made to derive SSA from satellite observations (Mary et al.,
2013; Xiong et al., 2018).
This paper presents a new retrieval algorithm to derive SGS, SPS, and SSA from satellite
observations. In a snow-atmosphere system, satellite observed TOA reflectances are affected
by numerous snow and atmospheric parameters. The parameters, which will be estimated in the
framework of the eXtensible Bremen Aerosol/cloud and surfacE parameters Retrieval (XBAER)
algorithm, will be called the target parameters. Other parameters, which the TOA reflectance
also depends on, will be called the model parameters. In the case of the XBAER algorithm, the
target parameters are SGS, SPS, and SSA, whereas the model parameters are aerosol loading,
cloud optical thickness, and gaseous absorption. Throughout the paper, SGS will be
characterized by an effective radius. Following Baum et al., (2011), the effective radius is
defined as $3V/(4A_p)$, where V and $A_p$ are the volume and average projected area, respectively.
As can be seen, in the case of a spherical particle, the effective radius is equal to the radius of
the sphere. The general concept of the retrieval algorithm is to use simultaneously spectral and
angular reflectance measurements, which are sensitive to SGS and SPS. The spectral channels
used in the XBAER algorithm are 0.55 μm and 1.6 μm. Both nadir and oblique observation
directions from SLSTR are used. An optimal SGS and SPS pair is achieved by minimizing the
difference between measured and simulated atmospheric-corrected surface reflectances. SSA
is then calculated based on the retrieved SGS and SPS. Nine predefined SPSs (aggregate of 8
columns, droxtal, hollow bullet rosettes, hollow column, plate, aggregate of 5 plates, aggregate
of 10 plates, solid bullet rosettes, column) (Yang et al., 2013, see Table 1) are used to describe
the snow optical properties and to simulate the snow surface reflectance at 0.55 and 1.6 μm at
two observation angles.
Three points we would like to emphasize to avoid misunderstandings between snow
science community and remote sesning community.
➢    **Usage the Yang et al (2013) database for ice crystal in the air (ice cloud) and on**

**the ground (snow).** The optical properties of ice crystals presented by Yang et al.,

(2013) have been widely used to study ice clouds. In recent publications, it has been

demonstrated that they can also be used for snow studies (Räisänen et al, 2015;

Pirazzini et al., 2015; Saito et al., 2019; Schneider et al., 2019; Pohl et al., 2020b). In

fact, the single-scattering properties of ice crystals in Yang et al., (2013) database are

determined solely by given particle size, shape, and refractive index. They can be

used to describe the optical properties of both snow particles and ice cloud particles

when the particle models represent the aforementioned optical/physical properties

(Saito et al., 2019; Personal communication with Dr. Saito).

➢    **Snow particle shape observed from field measurements and derived from**

**satellite observations**. For scientists working in a laboratory or on campaign-based

studies, the best way to get an image of snow is to use an X-ray microtomography or

confocal scanning optical microscope/scanning electron microscope (Hagenmuller et

al., 2016; Baker et al., 2019; Personal communication with Dr. Ian Baker). In a field

measurement and its related application areas (e.g., calculation of snow albedo), a

spherical shape assumption is widely used because it is easier to derive other snow

properties such as SSAs and snow albedo based on this assumption, compared to

other more complicated shapes (see Appendix). The assumption of spherical and non-

spherical shape has much less impact on the estimation of snow albedo compared to

the bidirectional reflection features of snow (Grenfel and Warren, 1999; Dumont et

al., 2010). Because SPS has a significant impact on the ice crystal phase function

while it has a relatively weak impact on the snow extinction/absorption coefficient (Jin et al., 2008). However, the spherical shape cannot be used to provide typical bidirectional reflection features of snow with required accuracy (Jin et al., 2008; Dumon et al., 2010; Jiao et al., 2019), which is the fundamental basis to derive snow properties from satellite remote sensing techniques. Thus, more complicated SPSs, such as those proposed by Yang et al (2013), are recommended to use in the simulations of the angular distribution of snow reflectance. Besides, both snow albedo and directional reflectance are affected by other factors such as how single particle aggregates;

➢ **SGS and SSA**. Although the definition of snow grain constitutes is an ongoing debate in different communities, SGS and SPS are two fundamental inputs for any radiative transfer model, which is the basis for the satellite retrievals (Langlois et al., 2020). Typically, the SSA is more preferable within the snow science community because SSA is commonly used in further applications based on field measurements. We note, however, according to the definition of SSA, for a given SPS, a unique relationship between SGS and SSA can be derived. SPS is the intermediate but fundamental parameter needed to retrieve SSA in our XBAER algorithm.

This paper is structured as follows: observations characteristics of SLSTR and the laboratory measurements used for sensitivity studies are described in section 2. The theoretical background and the ice crystal database (Yang et al., 2013) are presented in section 3. Section 4 describes the eXtensible Bremen Aerosol/cloud and surfacE parameters Retrieval (XBAER) algorithm. The results of a comprehensive sensitivity study using SCIATRAN (Rozanov et al., 2014) simulations are presented in section 5. The conclusions are given in section 6.

Table 1 Snow particle shape provided in Yang et al (2013) database. The abbreviations are introduced here will be used later

| Snow particle shape | Abbreviation | Schematic drawing |
|---|---|---|
| Aggregate of 8 columns | col8e |  |
| Droxtal | droxa |  |
| Hollow bullet rosettes | holbr |  |
| Hollow column | holco |  |
| Plate | pla_1 |  |
| Aggregate of 5 plates | pla_5 |  |
| Aggregate of 10 plates | pla_10 |  |
| Solid bullet rosettes | solbr |  |
| Column | solco |  |

**2 Data**

**2.1 SLSTR instrument**

The satellite data will be used twofold throughout the paper. In the first part, we perform a statistical analysis of the SLSTR observation/illumination geometries to select realistic settings for the sensitivity study. In the second part of the companion paper, the satellite measurements

will be used as the inputs of the XBAER algorithm to derive the research satellite products of
SGS, SPS, and SSA.
The SLSTR instrument onboard the European Space Agency (ESA) satellite Sentinel-3 is the
successor of the Advanced Along-Track Scanning Radiometer (AATSR) instrument, which is used
to maintain continuity with the (A)ATSR series of instruments. SLSTR takes the heritage of AATSR
instrument characteristics, especially the dual-viewing observation capabilities and wavelength
settings. In order to have a reasonable setting for observation/illumination geometries in the
sensitivity study, we perform a statistical analysis of the SLSTR observation geometries (solar
zenith angle, SZA, viewing zenith angle, VZA, relative azimuth angle, RAA), similar as Mei et al
(2020a). This analysis is essential because 1) it provides a realistic setting of
observation/illumination geometries in our sensitivity studies; 2) it helps us to have a complete
understanding of the observation/illumination related surface/atmospheric properties. Here the
definition of RAA has been harmonized with SCIATRAN (Rozanov et al., 2014), namely, RAA
value is equal to 0° under strict glint condition. The statistical analysis has been performed using
observations over Greenland during April and September 2017. April and September are reported
to be representativeness months of the Arctic (Mei et al., 2020a). Please be noted that these two
months are picked up to represent the SLSTR observation characteristic with typical solar
illumination angle, the change of underlying surface properties plays no role in such selection. Fig.
1 shows the frequency of SLSTR observation geometries. The upper panel shows the SZA with
SLSTR nadir and oblique observations for April and September. We can see that the SZA occurs
frequently with a value of 70° for selected months. The VZA and RAA for oblique observation mode
are typically around 55° and in a range of [110°, 170°], respectively. The observation geometries
for nadir observation show relatively large variabilities due to larger swath width compared to
oblique (1400 km vs 700 km). Larger SZA can be found especially at the edge of the swath. The
VZA and RAA for oblique observation mode are typically in ranges of [0°, 55°] and [70°, 140°],
respectively. According to the statistical analysis, a combination of SZA, VZA, RAA of 70°, 30°,
135° for nadir observation and 70°, 55°, 135° for oblique observation can be a reasonable setting
for the SLSTR observation geometries for the sensitivity study.

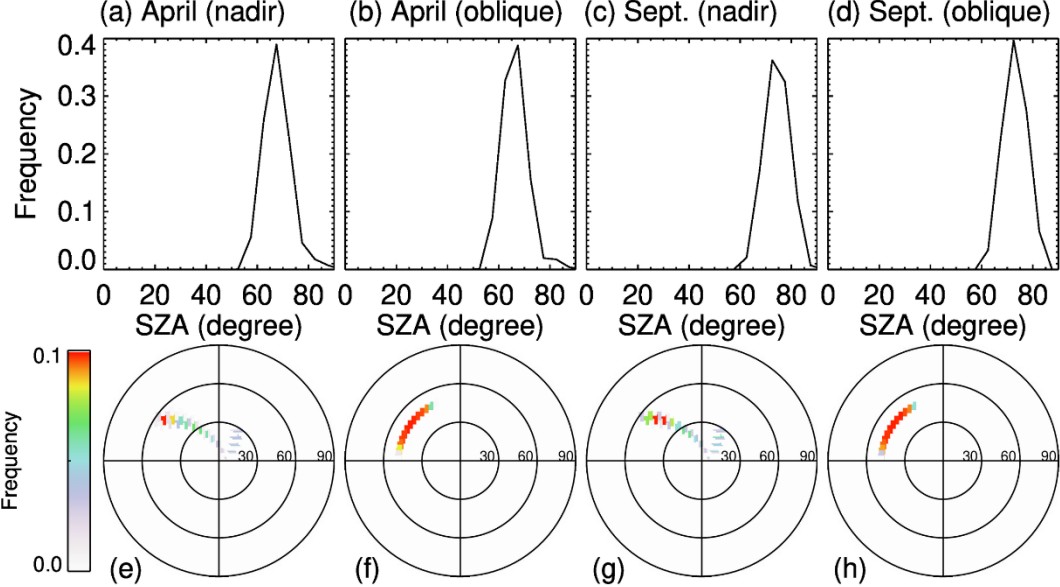


Fig. 1 Upper panel is the histograms of SZA for SLSTR observations: (a) nadir during April;
(b) oblique during April, (c) nadir during September; (d) oblique during September. Lower
panel is the polar plots of (VZA, RAA) probability for AATSR observations: (e) nadir during
April; (f) oblique during April, (g) nadir during September; (h) oblique during September.

## 2.2 Laboratory measurements

Laboratory measurements of the bidirectional reflectance of snow samples contain important
information about the dependence of the angular structure of snow reflection on the lighting
geometry, wavelength, and snow physical properties. The comparison of measured and
modeled bidirectional reflectance helps to establish the conceptual ideas for the retrieval
algorithm. For this comparison, we have selected measurements of fresh and aged snow
samples presented by Dumont et al., (2010) and Peltoniemi et al., (2009), respectively.
The fresh snow sample, a cylinder of 30 cm diameter and 12 cm height, was taken from
new wet snow layer at Col de Porte (Chartreuse, France) at 1300 meter above sea level during
January 2008 (Dumont et al., 2010). The sample was stored in a cold room at -10°C for one
week to avoid metamorphic effects during the ensuing measurements. To obtain the
Bidirectional Reflectance Factor (BRF), the snow sample was illuminated by a monochromatic
light source at incidence zenith angle of 60°. The spectral BRF between 500 and 2600 nm was
measured at viewing zenith angles of 0°, 30°, 60°, 70° and relative azimuth angles 0°, 45°, 90°,
135°, 180° by a spectrogonio-radiometer developed at the Laboratoire de Planétologie de
Grenoble, France, and using a Spectralon® and an infragold® sample as a reference (see
Dumont et al., (2010) for further details).

The aged snow sample, a cuboid of more than 10 cm height, was taken from an old dry

snow layer at Masala, Finland, and brought into a warm laboratory. The spectral BRF between
350 and 2500 nm was measured during the aged process by the Finnish geodetic institute field
goniospectro-polariphotometer (FIGIFIGO) and using a Labsphere Spectralon 99% white
reference plate. For illumination, a 1000 W Oriel Research Quartz tungsten halogen lamp at a
zenith angle of 60° was utilized (Peltoniemi et al., 2009). Spectral BRF was obtained at viewing
zenith angles up to 70° in 1° resolution and at relative azimuth angles of 0°, 90°, 130°, 160°,
180°, 270°, 310°, and 340°. The first and last measurements were done in the principal plane,
indicating minor metamorphism in the snow layer during the measurement.


**3 Dependence of snow reflectance on target parameters**
A comprehensive data library (Yang et al., 2013) containing the scattering, absorption, and
polarization properties of ice particles in the spectral range from 0.2 to 15 μm was used to
calculate radiative transfer through a snow layer (Pohl et al., 2020b). A full set of single-
scattering properties is available for nine ice crystal habits presented in Table 1. The maximum
dimension of each habit ranges from 2 to 10000 μm in 189 discrete sizes.

The optical properties of ice crystals depend on wavelength, ice crystal size, and shape.

Maximal dependence of the single-scattering albedo on the particle size is observed in the
spectral ranges where ice absorption cannot be neglected. The asymmetry factor depends on
the particle size for the whole spectral range. This dependence can be weaker or stronger at a
selected wavelength depending on SPS (see Yang et al., (2013) for details).

To better illustrate the impact of SGS and SPS on the radiative transfer through a snow

layer, we have calculated the reflectance of the snow layer consisting of droxtals, aggregates of
8 columns, hollow columns, and plates with crystal surface roughness condition as severely
roughened. The simulations of snow reflectance were performed using the radiative transfer
package SCIATRAN (Rozanov et al., 2014). The snow layer was defined as a layer directly
over a black surface, with snow optical thickness of 500 and a snow geometrical thickness of
1m. The snow layer is assumed to be vertically and horizontally homogeneous without any
surface roughness and composed of monodisperse ice crystals. The impact of snow impurities
and scattering/absorption processes in the atmosphere was neglected at this stage. The
reflectance of the snow layer as a function of the effective radius of ice crystal at wavelengths
0.55 μm and 1.6 μm is presented in Fig. 2. The calculations were performed for typical SLSTR
instrument observation/illumination geometries (see section 2.1), with SZA, VZA, and RAA
equal to 70°, 30°, and 135° (scattering angle 129°).
There are a couple of criteria we considered for the selection of the optimal wavelengths
(0.55 μm and 1.6 μm) in XBAER algorithm, for the purpose of creating a long-term satellite
snow properties dataset with good and stable accuracy.
➤    Taking the overlap channels between AATSR and SLSTR because a consistent
long-term satellite snow dataset is possible only when the same algorithm can be
applied on both AATSR and SLSTR instruments. In particular, the overlap channels
between AATSR and SLSTR are 0.55, 0.66, 0.87, 1.6, 3.7, 10.85, and 12μm.
➤    Picking up wavelengths, for which contribution of thermal emission can be
ignored, then 0.55, 0.66, 0.87, and 1.6 μm remain.
➤    Deleting the channel 0.66μm to avoid the potential impact of $O_3$ absorption,
after that, 0.55, 0.87, and 1.6 μm remain.
➤    Taking into account, that the retrieval algorithm is a two-stage algorithm, namely,
first it uses channels with minimum impact of ice crystal shape to retrieve the grain
size, and then it selects the shape using channels with minimum impact of grain size.
Accounting for that the channel 0.87μm is impacted by both size and shape, 0.55 and
1.6μm channels were picked up for the retrieval.
The right panel of Fig. 2 demonstrates the strong dependence of the snow layer reflectance
at 1.6 μm on the SGS. One can also see that the dependence of snow reflectance on SPS cannot
be neglected. In particular, the same reflectance can be obtained with a combination of different
SGS and SPS. For instance, one can see from the right panel of Fig. 2 that, the reflectance of
the snow layer consisting of droxtals with SGS=200 μm or of plates with SGS=65 μm equals
~0.035 in both cases. Thus, assuming different SPSs, the values of retrieved SGS can differ 3
times.The left panel of Fig. 2 demonstrates the dependence of the snow layer reflectance at 0.55
μm on SGS and SPS. It can be seen that the dependence of reflectance on SGS is very weak for
droxtals and aggregate of 8 columns. However, reflectance at 0.55 μm decreases with an
increase of SGS for hollow columns and plates. The weak oscillations for the reflectances at
0.55 μm can be explained by the joint impact of oscillations in the single-scattering albedo and
elements of the scattering matrix presented in the original database. Although the reason for the
oscillation in the database is unclear, it is unlikely due to physical phenomena (Dr. M. Saito ,
personal communication).

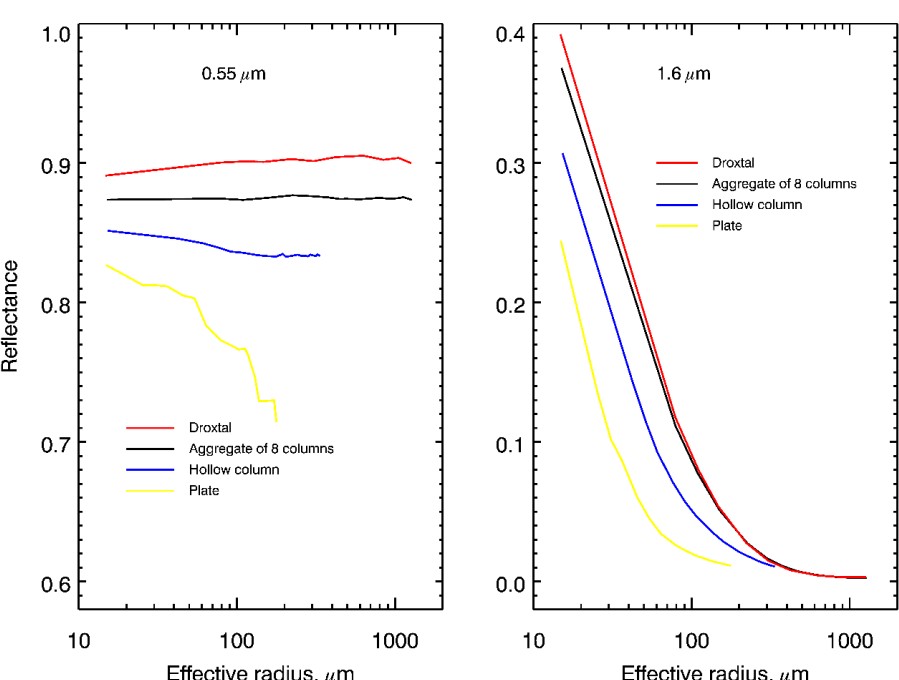


Fig 2. Reflectance of snow layer at 0.55 μm and 1.6 μm calculated assuming different SPS.
Observation/illumination geometry: SZA, VZA and RAA were set to 70°, 30° and 135°,
respectively.
To illustrate this point, the dependence of the phase function at 129° scattering angle on
SGS is shown in the left panel of Fig. 3. The phase functions (F11 element of the scattering
matrix) were extracted from the original database. According to the left panel of Fig. 3, the
dependence of snow surface reflectance at 0.55 μm on SGS and SPS is caused mainly by the
phase function of ice crystal. Weak oscillations can also be found.
The above analysis shows that accurate retrieval of SGS requires adequate information
about SPS and accounting for the dependence of the phase function on SGS. To better illustrate
the impacts of SGS on ice crystal phase function, we calculated reflectance at 1.6 μm with
different SGS values. The right panel of Fig. 3 represents the reflectance of the snow layer,
consisting of aggregates of 8 columns, calculated accounting for the dependence of the phase
function on the effective radius (black line) and assuming constant phase function for three
selected effective radii equal to 15, 150, and 1150 μm (red, green, and blue lines, respectively).
It can be seen that the accurate simulation of snow reflection requires accounting for the
dependence of phase function on SGS.
The main findings of presented investigations can be formulated as follows:
➢    reflectance of a snow layer depends on both SGS and SPS;
➢    accurate simulation of snow surface reflectance requires accounting for the dependence of

phase function on SGS;

➢    spectral channels in the visible spectral range is more sensitive to SPS compared to SGS;
➢    spectral channels in the near infrared spectral range is more sensitive to SGS compared to

SPS.


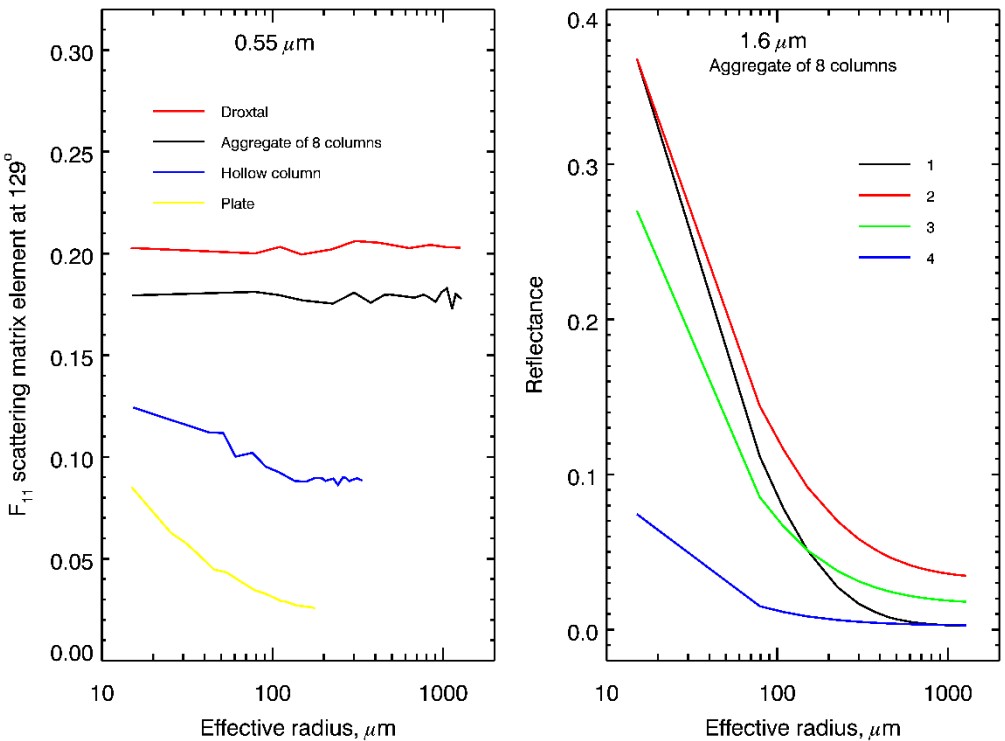


324 Fig 3. Left panel: phase function at 0.55 μm for scattering angle of 129°, extracted from the

325 original database (Yang et al., 2013) as a function of effective radius. Right panel: reflectance

326 of snow layer at 1.6 μm consisting of aggregate of 8 columns, calculated assuming that: 1:

327 phase function depends on the effective radius (black line); 2: phase function is constant

328 corresponding to the effective radius 15μm (red line); 3: same as 2 but for effective radius of

329 150 μm (green line); 4: same as 2 but for effective radius of 1150 μm (blue line).

331   Although the global classification snow crystal, ice crystal, and solid precipitation

332 particles suggested in Kikuchi et al. (2013) consist of the 121 particle types, we restrict

333 ourselves, in the retrieval algorithm, with nine shapes of ice crystals, for which optical

334 characteristics are represented in database (Yang et al., 2013). And these nine shapes have been

335 proven to be used to reproduce typical wavelength/angular features of snow reflectance in

336 reality, especially from satellite observations (Räisänen et al, 2015; Pirazzini et al., 2015; Saito

337 et al., 2019; Schneider et al., 2019; Pohl et al., 2020b). To futher illustrate that the selected

338 dataset is able to reproduce the BRF of different snow types, we compared the simulated and

339 measured BRF of fresh (Dumont et al., 2010) and aged (Peltoniemi et al., 2009) snow samples.

To reproduce the spectral BRF by SCIATRAN, we use the setup described above in this section
and adjust the SGS for each SPS by minimizing the deviation between simulated and measured
reflectance at 1.6 μm. Figure 4 shows the simulated BRF in the principal plane at 0.55 μm of
fresh and aged snow samples, as well as the respective measurements. The BRF is defined as
πI/F, where I is the reflected radiance and F is the incident irradiance. According to Fig. 4(a),
for fresh snow, plates are the best shape to reproduce the measured BRF in the vicinity of the
forward scattering peak but plates underestimate the BRF at higher viewing zenith angles in the
backscattering region. Here, shapes of hollow bullet rosette, hollow column, aggregate of 10
plates exhibit better potential to simulate the fresh snow layer BRF. In the case of aged snow,
shapes of solid and hollow column, hollow bullet rosette, and aggregate of 5 and 10 plates
provide BRF values in conformity with respective measurements. However, they slightly
underestimate the BRF at high zenith angles in the backscattering region where aggregate of 8
columns can simulate the aged snow BRF better.

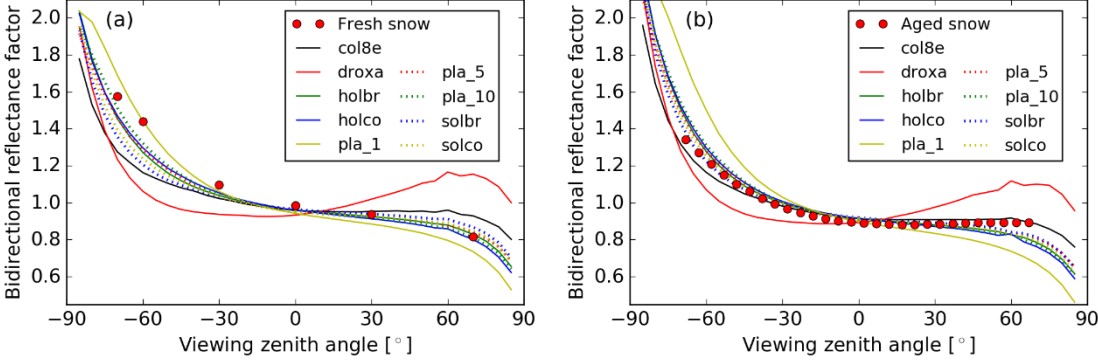


Fig. 4 The comparison of angle dependence of laboratory-measured and simulated snow
reflectance: (a) fresh snow sample; (b)aged snow sample. Symbols - measurements, lines –
simulations with SCIATRAN assuming different SPS (see legend).
The above analysis demonstrates that the selected database of SPS can be used
successfully to reproduce measured BRF of both fresh and aged snow samples. Similar results
were obtained by Pohl et al., (2020b). In this paper, top of atmosphere BRF at 865 nm derived
from  POLarization and Directionality of  the Earth's Reflectances 3 (POLDER-3) on
Polarization & Anisotropy of Reflectances for Atmospheric Sciences coupled with
Observations from a Lidar (PARASOL) measurements over a pure snow surface in Greenland
(70.5° N, 47.3° W) on 6 July 2008 were compared with the SCIATRAN simulations, using
droxtals, solid bullet rosettes, and solid columns.
According to the above analysis, we can formulate the general algorithm to retrieve SGS
and SPS from satellite observations. Satellite provides the wavelength-dependent TOA
reflectance, for a given SGS and SPS pair, the minimization between satellite observed TOA
reflectance and theoretical simulation is performed. The optimal SGS and SPS are obtained
when the difference between observations and simulations reaches the predefined criteria. The
SSA is then calculated by the retrieved SGS and SPS.

## 4 XBAER Algorithm

The retrieval algorithm consists of three stages. The first stage includes the estimation of
SGS using the effective Lambertian surface albedo after atmospheric correction for selected
observation geometries and wavelengths. This step is performed based on the path radiance
representation (Mei et al., 2017), in which the TOA reflectance can be described by the
contribution from the atmosphere and the interaction between atmosphere and surface. The
inverse to derive the surface reflectance from the satellite observed TOA reflectance is called
the atmospheric correction. And due to certain assumptions in the path radiance
representation, the derived surface reflectance is equivalent to the effective Lambertian
surface albedo. The estimation of SGS is obtained solving the following minimization
problem with respect to the effective radius, $r$, of snow crystals:
$$\left\| \boldsymbol{A}_e - \boldsymbol{R}_s(r) \right\|^2 \to \min. \tag{1}$$

Here, $\mathbf{A}_e$ and $\mathbf{R}_s(r)$ are two vectors which components are the effective Lambertian
surface albedo and the simulated snow reflectance, respectively. The dimension of these
vectors is the number of wavelengths times the number of viewing directions.
The simulation of snow reflectance (components of vectors $\mathbf{R}_s(r)$) was performed using
the radiative transfer package SCIATRAN (Rozanov et al., 2014) as described in Section 3.
The optical properties of nine SPSs, listed in Table 1, were used for radiative transfer
calculations.

The minimization problem formulated by Eq. (1) was solved separately for each SPS

using Brent's method (Brent, 1973). The solution of the minimization problem for each
crystal habit is characterized by the following residual:
$\Delta_i = \left\| A_e - R_s(r_i^*) \right\|^2, i = 1, 2, ..., 9,$                    (2)
where $r_i^*$ is the solution of minimization problem given by Eq. (1) for $i^{th}$ shape of the ice
crystal particle.

The second stage is the selection of such $i$ (SPS) for which $\Delta_i$ is minimal. This

completes the retrieval process and enables the optimal SGS and SPS to be obtained.

The third stage is to calculate SSA for the retrieved SGS and SPS. To this end, let us

rewrite the SSA introduced above in the following equivalent form:

$SSA = 3/\rho r \cdot (A_t/4A_p),$                    (3)

where $r$ is the effective radius. According to Cauchy's surface area formula (Cauchy, 1841;
Tsukerman and Veomett, 2016), the average area of the projections of a convex body is
equal to the surface area of the body, up to a multiplicative constant. In our case, this results
in $A_t = 4A_p$ and SSA for convex particles such as droxtals, solid columns, and plates are
equal to $3/\rho r$. In the case of non-convex particles, the calculation of SSA requires the
information about total area $A_t$. Although the database given by Yang et al. (2013) does not
contain information about $A_t$, the total area of non-convex particles can be calculated
employing geometric parameters of ice crystal habits presented in Table 1 of Yang et al.
(2013). Here we take a typical SPS, aggregate of 8 columns, as an example, to show the
difference between SSA calculated assuming convex and non-convex particle.

According to M. Saito (private communication), the parameters $L$ and $a$ of the

aggregate of 8 columns (see Fig. 3 in Yang et al (2013) for details) can be obtained by
scaling with respect to the maximum dimension, $D$. To find these values for different
maximal dimensions, we calculate at first the volume of aggregate of 8 columns
corresponding to parameters $a$ and $L$ on a relative scale as given in Table 1 of Yang et al

(2013).

$$V_r = \frac{3\sqrt{3}}{2} \sum_{i=1}^{8} a_i^2 L_i. \tag{4}$$

Using the database of Yang et al (2013), one can obtained the maximal dimension, $D_r$,
corresponding to the volume, $V_r$. Introducing the scaling factor, $C_k = D_k/D_r$, we have semi-
width and length for the aggregate with the maximal dimension $D_k$:
$$a_{i,k} = a_i C_k, \quad L_{i,k} = L_i C_k. \tag{5}$$

The total surface of the aggregate on relative scale is given by
$$S_r = 3 \sum_{i=1}^{8} (\sqrt{3} a_i^2 + 2 a_i L_i). \tag{6}$$

Accounting for Eq (5), we have
$$S = C_k^2 S_r. \tag{7}$$

Having obtained the total area, one can calculate SSA as the total surface area of a
material per unit of mass:
$$SSA = \frac{S}{\rho V} = \frac{S_r}{\rho C_k V_r}. \tag{8}$$

Comparing SSA of convex particle equal to $3/\rho r$ with result given by Eq. (8), one can
easily notice the difference of SSA calculated from different SPS using the same SGS. The
details of such calculations for other non-convex ice crystal habits are given in the Appendix.

The relationship between SSA and SGS for different SPS is presented in Fig. 5. According

to Fig. 5, an almost inverse linear relationship between SSA and SGS can be found. The lines,
representing droxtal, plate, and column, are overlapped, indicating the same SSA for convex
particles. For other SPSs with the same SGS, SSA is larger compared to convex faceted
particles. SSA is restricted in the range of 0-100 $m^2$/kg in this investigation (Picard et al.,
2009).For example, for SGS=100$\mu$m, the SSA is 32.7 m2/kg for convex faceted particles,
whereas SSAs for aggregate of 8 columns, hollow bullet rosettes, hollow column, aggregate of
5 plates, aggregate of 10 plates, and solid bullet rosettes are 44.2, 43.4, 37.7, 74.4, 66.8 and
35.6 $m^2$/kg, respectively. The relative differences range from 9%-128%, depending on the SPS.
Taking into account the definition of SSA, one can derive the following relationship between
SSA convex and non-convex particles: $SSA_{nc} = SSA_c \cdot (A_t/4A_p)$, where subscript c and nc
denotes convex and non-convex particle, respectively. The obtained results reveal that for all
non-convex ice crystals under consideration $A_t/4A_p > 1$ and the ratio $A_t/4A_p$ weakly depends
on the SGS.


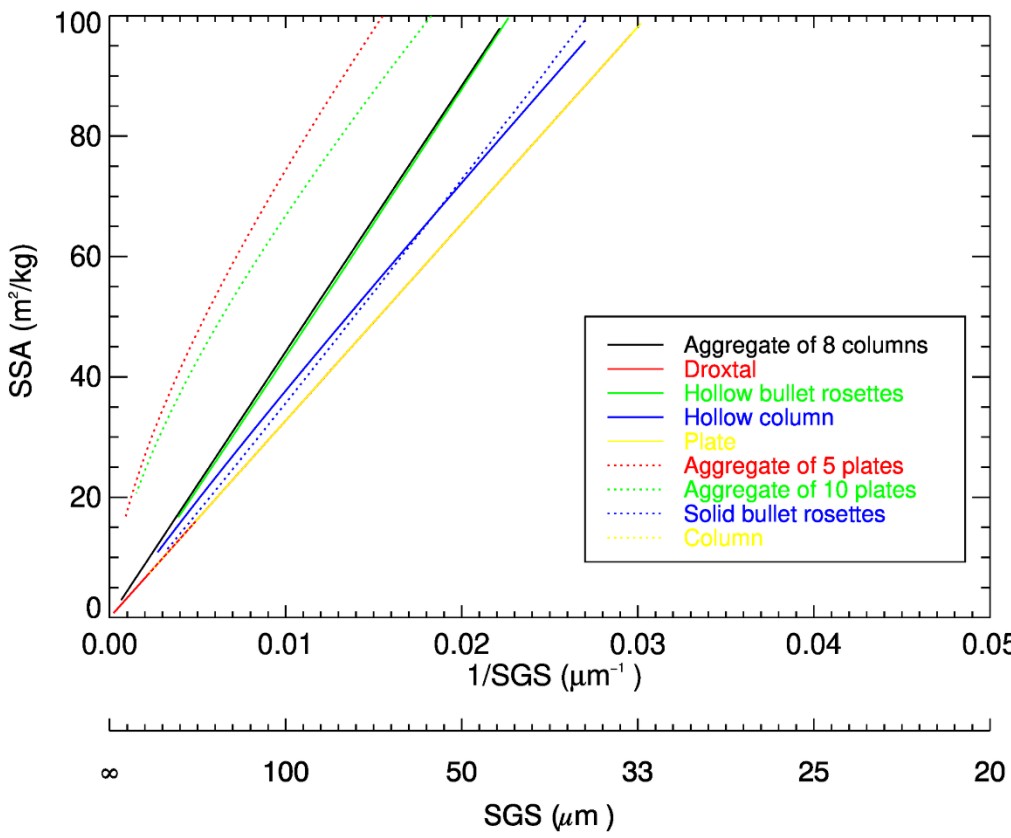


Fig 5. Relationship between SGS and SSA for different SPS. For a better illustration, the realistic range of specific surface area is limited to 100 m²/kg.

## 5 Impact of model parameters uncertainty

The accuracy of any retrieval algorithm depends not only on measurement errors but also on the uncertainty of parameters which cannot be retrieved. In our case, such parameters are ice crystal roughness, aerosol, and cloud contamination. The impacts of these factors on XBAER-derived SGS and SPS have been investigated and will be discussed in this section. The TOA reflectances at selected channels (0.55 and 1.6 μm) and observation directions for SZA, VZA, and RAA of 70°, 30°, and 135° for nadir 70°, 55°, and 135° for oblique, respectively, were calculated using radiative transfer model SCIATRAN. The details of each scenario will be presented in the corresponding sub-section below.

## 5.1 Impact of snow particle shape

Since the first stage of the XBAER algorithm is to estimate the SGS assuming a given SPS, it is reasonable to investigate the impact of SPS on the retrieval of SGS. The TOA reflectances of a snow layer at 0.55 and 1.6 μm with above-given observation geometries were calculated using the following settings for snow layer and atmospheric parameters:

➢ **Snow Layer:** consists of ice crystals with SPS set to be severely roughened aggregate of 8 columns and maximal dimensions [100, 300, 500, 700, 1000, 2000, 3000, 5000] μm, which corresponds to SGS [15, 45.1, 75.2, 105.3, 150.4, 300.8, 451.3, 752.1] μm.

➢ **Atmosphere:** excluded

The simulated snow reflectances were used as components of vector $\mathbf{A_e}$ in Eq (1). Nine SPSs from database presented in Yang et al. (2013) are used sequentially in the retrieval process. The atmospheric correction is not performed because the atmosphere is excluded in the forward simulations. This enables avoiding additional errors caused by the atmospheric correction and estimates the pure effect of SPS on the retrieval results. Fig.6 shows the impact of the SPS on SGS retrieval. Different colors and line styles indicate different ice crystals used in the retrieval process. The black solid line represents the retrieved SGS assuming SPS in the retrieval process is the same as in forward simulations. This line agrees well with the 1:1 line, indicating that the retrieval algorithm has been implemented technically correct. According to Fig.6, one can see both underestimation and overestimation of SGS depending on the SPS used in retrieval. However, in most cases, an incorrect SPS leads to an underestimation of SGS. In particular, the maximal effect can be seen when ice crystals of plate shape, rather than the correct aggregate of 8 columns, is used (yellow solid line). This result can be easily explained coming back to the right panel of Fig. 2. Indeed, one can see that the same reflectance of the snow layer can be obtained using the plate shape, instead of an aggregate of 8 columns, with significantly smaller SGS. These results reveal that the SPS is an important parameter affecting the accuracy of retrieved SGS.

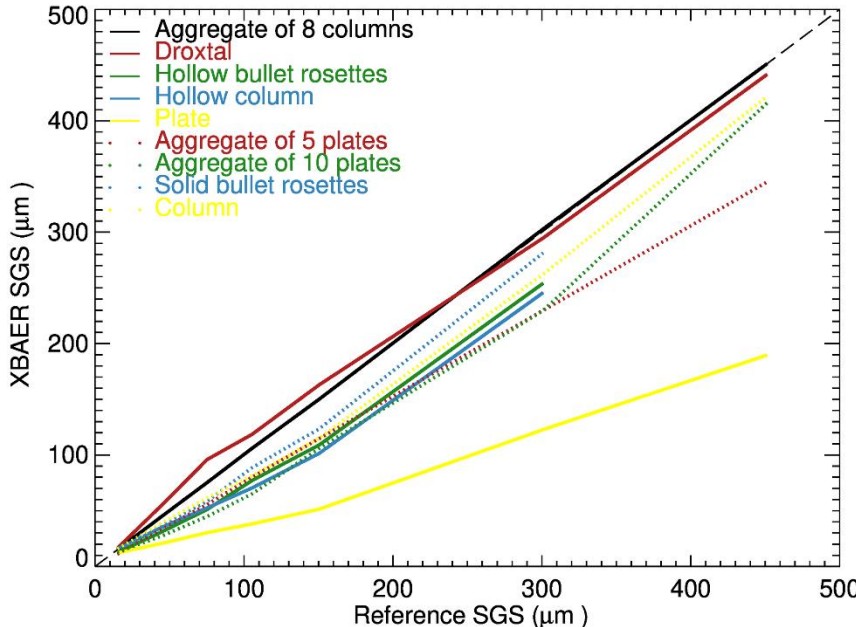

Fig 6. Impact of SPS on the retrieval of SGS.

## 5.2 Impact of SGS/SPS on SSA

Since the SSA is obtained from the retrieved SGS and SPS, an understanding of how the error of SGS and/or SPS propagates to the SSA will provide helpful information to understand the retrieved SSA. Fig. 7 shows the impact of SGS (left ) and SPS (right) on XBAER retrieved SSA. The relative error of SGS, $\varepsilon_r = (r - r')/r$, is propagated to the relative error of SSA as $\varepsilon_{SSA} = 1 - 1/(1 - \varepsilon_r)$, and it is independent of reference SSA. The left panel of Fig. 7 depicts $\varepsilon_{SSA}$ corresponding to $\pm 0.16$ of $\varepsilon_r$. One can see that this results in 19% and -13.8% of SSA relative errors, which are presented as the upper and lower error boundaries in the left panel of Fig. 7. The systematical error of ±16% for SGS was obtained as the maximal relative difference between XBAER retrieved SGS and both *in-situ* and aircraft measured SGS (as presented in the companion paper). This represents the worst case of SGS error propagation into SSA.

The impact of SPS on SSA is demonstrated in the right panel of Fig. 7. As a reference shape, we have selected in this case the plate, which provides the same SSA as other convex particles. One can see that the SSA of non-convex particles overestimates the SSA of convex particles, which is in line with the results presented in Section 4. For instance, for the same

SGS, the SSA for aggregate 8 columns (non-convex particle) is about 3 times larger than that
for doxtal (convex particle). Since the assumption of the sphere (convex particle) is used to
measure SSA in-field measures (Gallet et al., 2009; Personal communication with Dr. Nick
Rutter), such as observations from SnowEx, the retrieval results of SSA from XBAER will be
systematically larger than field measurements in the case of non-convex particles even if the
retrieved and measured SGS are similar. However, a detailed discussion with respect to
uncertainty in the campaign-based measurement is out of the scope of this manuscript.

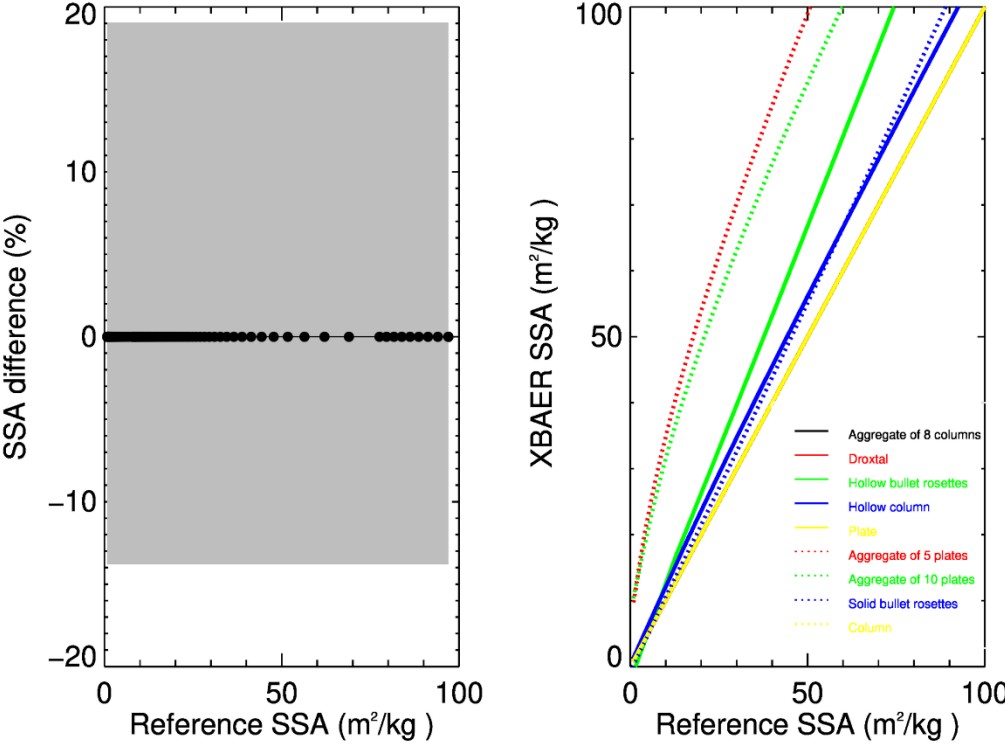


Fig 7. Impact of SGS and SPS on the retrieval of SSA. Left panel (SGS errors): the black line
with dots indicate the 0 difference for accurate SGS for aggregate 8 column, the grey area
indicate the relative error of SSA introduced by 16% error of SGS; Right panel (SPS selection):
different color/line styles indicate different SPS used in the calculation of SSA while the true
SPS is set to be „ plate" or other convex particles.


## 5.3 Impact of ice crystals surface roughness

Although surface roughness of ice crystal is not so severe for snow compared to ice cloud due to basic thermodynamics (Colbeck, 1980, 1983), the Ice Crystal Surface Roughness (ICSR), indicating ice crystal surface texture, may still be important for the retrieval of snow properties from optical sensors such as SLSTR. The ICSR has been used as a new variable in model simulation (Järvinen et al., 2018). Retrieval algorithms of ice cloud parameters frequently based on the assumption that the ice crystal surface is smooth (Kokhanovsky et al., 2019). This assumption can yet introduce large uncertainty in the ice cloud retrieval parameters and, as a consequence, lead to misunderstanding the impacts of ice cloud on global climate change (Järvinen et al., 2018). However, this issue has not yet been discussed for snow. In general, ice crystal surfaces are rougher in clouds than in snow layers due to metamorphism processes (Colbeck, 1980, 1983; Ulanowski et al., 2014). The investigation of the impact of ICSR on retrieval of snow properties provides valuable information to understand the XBAER algorithm. The ICSR according to Yang et al., (2013) is defined similarly as suggested by Cox and Munk (1954) for the roughness of the sea surface. A parameter $\sigma$ describes the degree of ICSR. The $\sigma$ values 0, 0.03, and 0.5 are for three surface roughness conditions: smooth, moderate roughness, and severe roughness. And only the above three values are available in the Yang database. The snow layer reflectances were used as components of the vector $\mathbf{A}_e$ in Eq. (1) in the same way as in Section 5.1.

Fig. 8 shows the impact of ICRS on the retrieved SGS, SPS, and SSA. The impact of ICRS on SGS and SSA are relatively small for SGS smaller than ~300 μm. Ignoring the impact of roughness leads, in general, to a slight overestimation on SGS and an underestimation of SSA. The absolute errors of SGS and SSA introduced by ICRS range from 0.3% - 3%, depending on SGS. Due to the inverse almost linear relationship between SSA and SGS, as presented in Fig. 5, for the same SPS, an overestimation of SGS leads to an underestimation of SSA. The slight overestimation can be found if less ICRS is taken into account in retrieval because the snow reflectance with the same SGS and SPS for ICRS = 0.5 is larger than for ICRS = 0.03 due to lower asymmetry factor of ice crystal with more roughened surface roughness, thus the same surface reflectance observed by satellite requires larger SGS for the case with ICRS = 0.03 used

in retrieval in contrast to ICRS = 0.5 used in the forward simulation. However, as can be seen
from the right panel of Fig.8, the XBAER algorithm still retrieves the correct SPS ignoring the
impact of roughness.

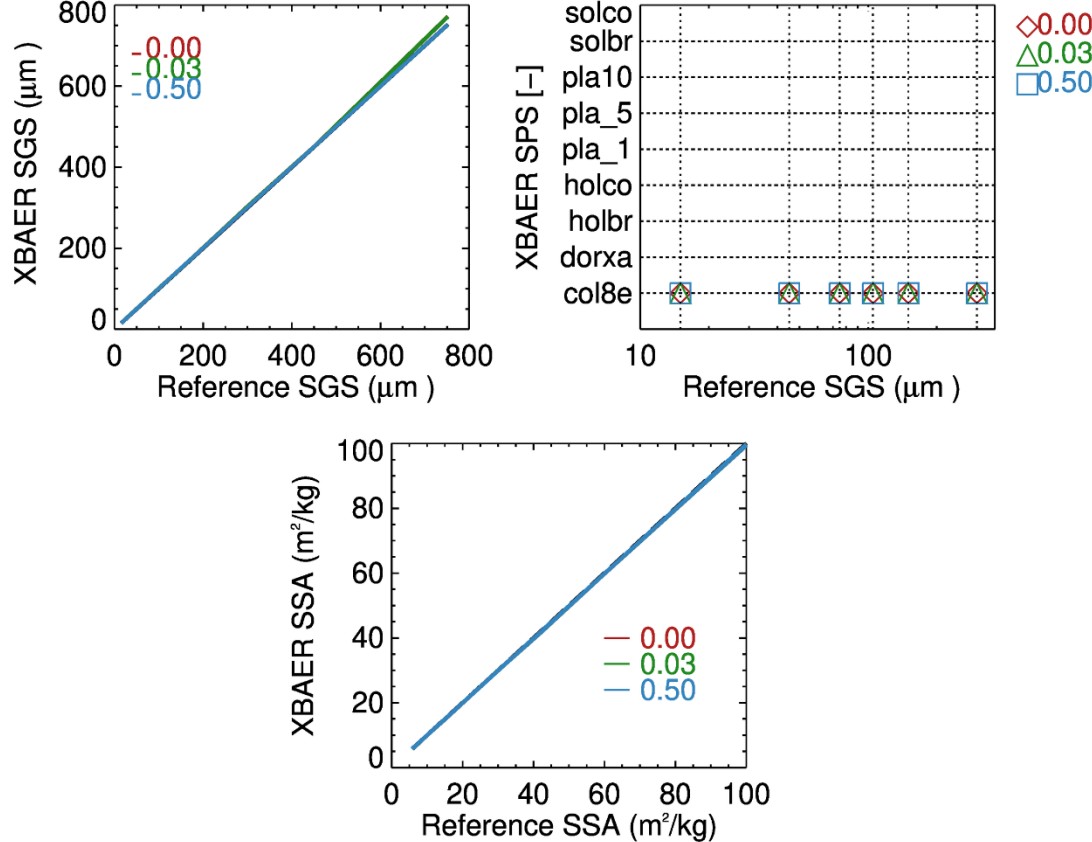


Fig 8. Impact of Ice Crystal Surface Roughness (ICSR) on the retrieval of SGS (upper left)
SPS (upper right) and SSA (lower). Different colors indicate different ICSR used in the
retrieval.

## 5.4 Impact of aerosol contamination

The impact of aerosol on the retrieval of snow properties using passive remote sensing can be
important because there is limited aerosol information over the cryosphere (Mei et al., 2013a;
Mei et al., 2013b; Mei et al., 2020a; Tomis et al., 2015) to perform an accurate atmospheric
correction. The use of MERRA simulated AOT, although with good data quality, will still
introduce potential aerosol contamination in the XBAER-derived snow properties. The impact
of aerosol on snow properties retrieval is much smaller over Arctic regions compared to middle-
low latitude (e.g. Canadian Arctic, Tibetan Plateau) due to large absolute uncertainty in the
MERRA simulated aerosol over middle-low latitude in wintertime. A detailed comparison of
how possible aerosol contamination may affect the retrieved snow properties will be included
in the companion paper (Mei et al., 2020c). In the companion paper, the comparison between
satellite-derived and campaign-measured snow properties all over the world will be included.
In order to have a better understanding of aerosol contamination on snow properties retrieval,
the TOA reflectances were calculated at 0.55 and 1.6 μm with above-given observation
geometries using the following settings:
➤  **Snow Layer:** Same as in section 5.1;
➤  **Atmosphere:**
●  Aerosol type is set to be weakly absorbing (Mei et al., 2020b) with AOTs [0.05, 0.08,0.11].

Other atmospheric parameters are set according to Bremen 2D Chemical transport model

(B2D CTM) for April at 75° N (Sinnhuber et al., 2009). It is worth to notice this three AOT

values represent background, average, and pollution conditions in the Arctic as suggested

by Mei et al (2020a; 2020b).

Fig.9 shows the impact of aerosol contamination on the SGS (upper left), SPS (upper right),
and SSA (lower) retrieval. These results are obtained by introducing 50% error in AOT at the
step of atmospheric correction and can be considered as the worst case for impact of aerosol
contamination on retrieved SGS, SPS, and SSA. The surface reflectances estimated after
employing the atmospheric correction were used as components of the vector $\mathbf{A_e}$ in Eq. (1). One
can see that aerosol introduces systematic underestimation of retrieved SGS for the given
scenarios and the magnitude of underestimation increase with the increase of AOT. For a typical
background Arctic aerosol condition, with AOT=0.05, aerosol contamination introduces errors
in SGS of less than 3% for SGS ≤150 μm, and less than 7% for 150≤SGS<300 μm. The
maximal errors introduced by the aerosol contamination increase to 30% and 37% in the case
of average and pollution conditions for AOT=0.08 and 0.11, respectively. Please be noted that
the AOT values in the Arctic can be even smaller than 0.05, for instance, AOT over Greenland.
Thus, the analysis with respect to aerosol contamination is the worst case for a typical Arctic
condition.
For the case of AOT = 0.05, SPSs have been correctly retrieved for all SGS values,
indicating that under a typical Arctic clean condition, the impact of aerosol is not so large to
disturb SPS retrieval. In order to demonstrate the two stages retrieval process and illustrate the
impact of aerosol, let us focus on Fig. 10. To facilitate the presentation, we consider the
measurement of reflectance at 1.6 μm for a single observation direction (30°) and at 0.55 μm
for the difference of reflectance at two observation angles (30° and 55°). This enables avoiding
the minimization process given by Eq. (1) and represents the retrieval process in the simple
graphic form. The left panel of Fig. 10 depicts the determination of an effective radius for each
ice crystal form, assuming the correct shape is aggregate of 8 columns with an effective radius
105.4 μm. Solid and dotted lines are surface reflectance of the snow layer consisting of ice
crystals with different forms and the dashed line is the measured reflectance after the
atmospheric correction. The obtained SGSs are in the range 40 – 120 μm, depending on the
selected SPS, and presented in Fig. 10 by solid and dotted vertical lines. In the case of correct
SPS selection (aggregate of 8 columns) the retrieved SGS is ~110 μm. The right panel of Fig.
10 shows the second stage of the retrieval process, namely, the selection of such SPS for which
the difference between measured (dashed line) and simulated value (solid black line) is minimal.
In the case under consideration the correct shape is selected with an effective radius ~110 μm.
For larger AOT conditions, an inaccurate selection of SPS occurs for all SGS cases,
indicating the remaining aerosol information is large enough to decouple the aerosol
contribution from the snow surface characteristic. Thus, a quality flag of SPS, associated with
AOT, should be introduced in the retrieval of real satellite data. It is interesting to see that "solid
bullet rosettes" is the preferable SPS for very strong aerosol contamination cases. This is due
to similar scattering properties (shape) of ice crystal and weakly absorbing aerosol, defined in
forward simulation. The impact of aerosol contamination, for typical Arctic conditions,
introduces less than 5% error in SSA. However, for large aerosol contamination, the around 30%
underestimation in SGS linearly introduced about 25% overestimation in SSA, which agrees
with the analysis as presented in Fig.7.

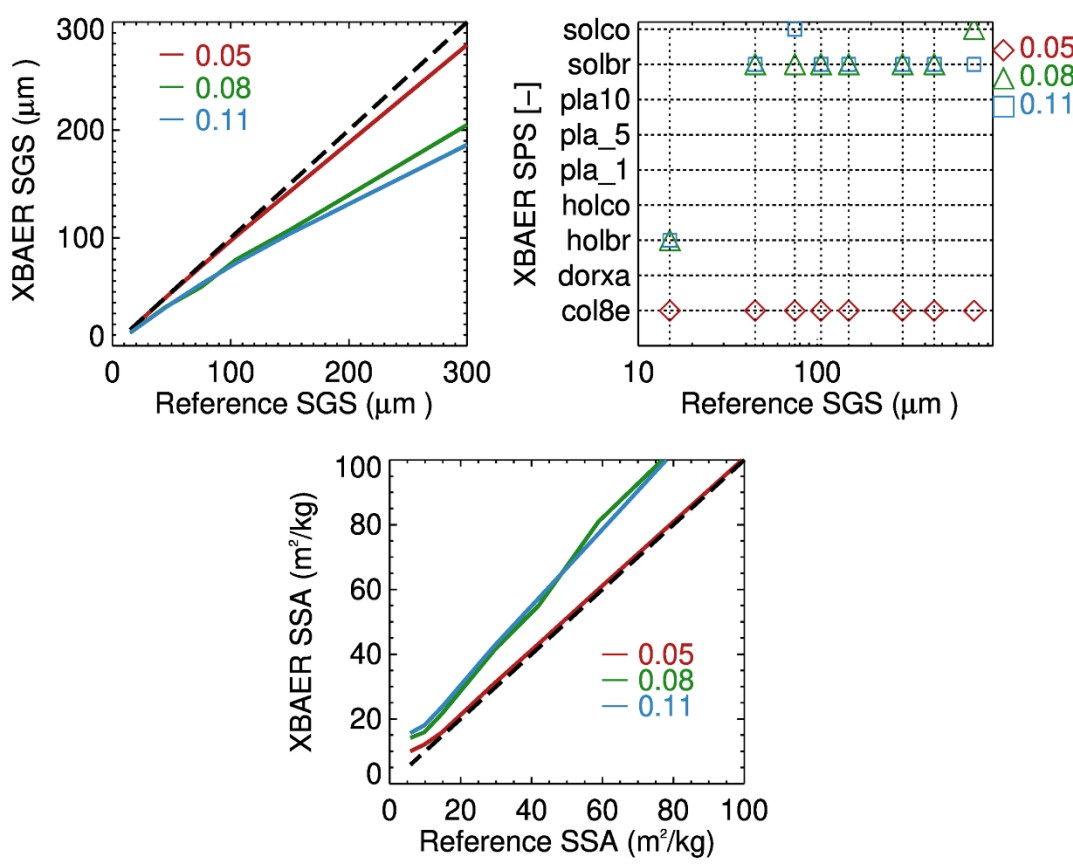


Fig 9. Impact of aerosol contamination on the retrieval of SGS (upper left) SPS (upper right) and SSA (lower). Different colors indicate different AOT used in forward simulations. No atmospheric correction is performed in the retrieval, black dash line is the 1:1 line.


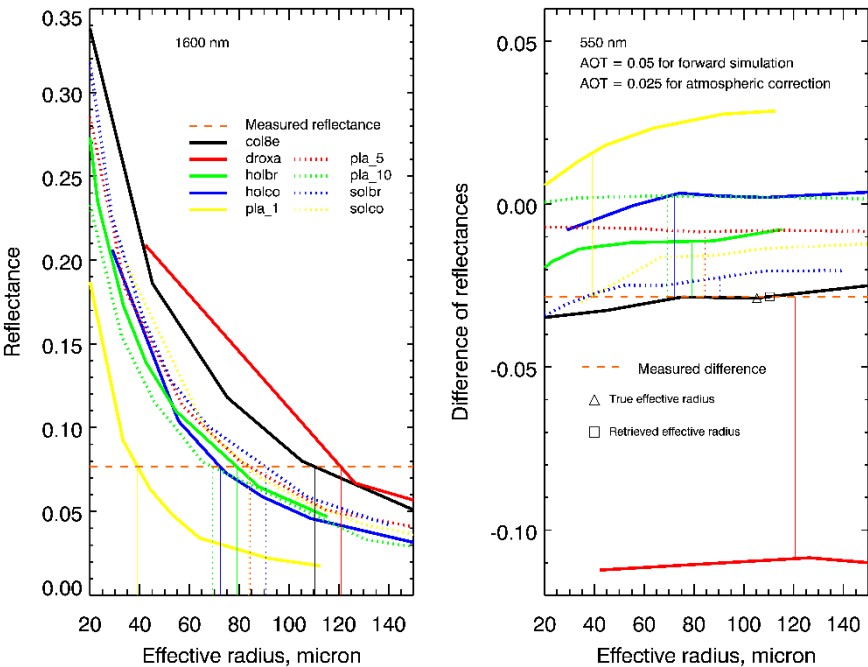


Fig 10 Schematic representation two stages of the retrieval process. Left panel: determination of effective radius for each ice crystal form. Right panel: selection of optimal SGS, SPS pair.

## 6 Impact of cloud contamination

Any cloud screening method, especially over the cryosphere, may introduce cloud contamination for the retrieval of atmospheric and surface properties (Chen et al., 2014; Mei et al., 2017; Jafariserajehlou et al., 2019). Understanding of the cloud contamination will provide valuable information to interpret the retrieval results using the SLSTR instrument. To investigate the impact of cloud contamination, the following settings were used to perform the simulations of TOA reflectance:

➤ **Snow Layer:** Same as section 5.1;

➤ **Atmosphere:** Aerosol free atmosphere with other parameters as in section 5.4. Additionally, vertically homogeneous ice cloud consisting of aggregate of 8 columns with effective radius of 45 μm and optical thickness [0.1, 0.5, 1.0, 5] is set to be at position of [5 km, 6 km].

Fig. 11 shows the impact of cloud contamination on XBAER retrieved SGS (upper left), SPS (upper right), and SSA (lower). The size of ice crystals in ice clouds is typically smaller than snow grain size (Kikuchi et al., 2013). Our statistical analysis of ice crystal effective radius

648 over Greenland shows an average value in the range of 30-50 μm, which is consistent with

649 previous publications (King et al., 2013; Platnick et al., 2017). According to Fig.11, an

650 overestimation of SGS can be found for SGS less than 45μm (cloud effective radius) and an

651 underestimation of SGS for SGS larger than 45μm. The magnitude of

652 overestimation/underestimation increases with the increase of Cloud Optical Thickness (COT).

653 XBAER derived SGS becomes saturated for COT larger than 0.5. Due to limited photon

654 penetration depth for optically thicker clouds (e.g., COT = 5), the XBAER algorithm retrieves

655 the effective radius of ice crystal in the cloud. This demonstrates that theoretically, the XBAER

656 algorithm can retrieve an ice cloud effective radius without a pre-processing of cloud screening.

657 And this can be further used as post-processing to avoid cloud contamination.

659  The impact of the cloud on the retrieval of SPS is similar to the impact of aerosol

660 considered above. In short, the cloud plays a larger role for larger SPS (darker TOA) and this

661 impact increases with the increase of COT. However, cloud with large COT can be much easier

662 detected and excluded by the cloud screening algorithm (e.g for the cases with COT > 0.5).

663 SPSs are correctly picked up due to the same SPS used for both the snow layer and the cloud

664 layer. Similar to the impact of aerosol, the underestimation of SGS introduced by the cloud

665 leads to an overestimation of SSA (Fig. 11 (lower panel)). The increase of COT results in

666 saturation of the ice cloud SSA, with a value of 100 $m^2$/kg in the case of aggregate of 8 columns.

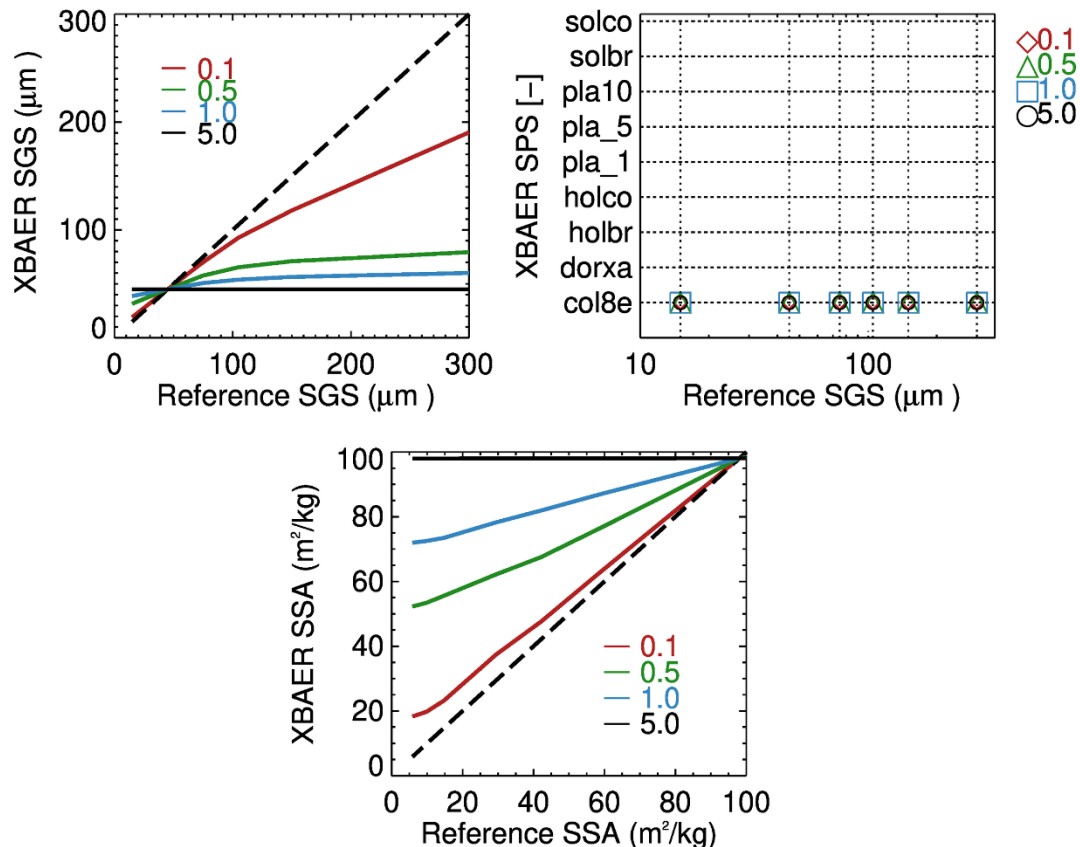

Fig 11. Impact of cloud contamination on the retrieval of SGS (upper left) SPS (upper right) and SSA (lower). Different colors indicate different COTs in forward simulations, black dash line is the 1:1 line.

## 7 Impact of other factors occurring in reality

The above theoretical investigations include all possible important factors affecting the accuracy XBAER algorithm. However, when applying XBAER algorithm to the SLSTR instrument for real scenarios, two additional factors need to be considered as well. One is the impact of the instrument spectral response function (SRF), the other one is the representativeness of the snow scenario for reality.

### 7.1 Impact of instrument spectral response function

➢ **Snow Layer:** Same as section 5.1;

➢ **Atmosphere:** Aerosol free atmosphere with other parameters as in section 5.4.

684 The forward simulations are performed with and without the impact of Spectral Response

685 Function (SRF). The SRFs for SLSTR at 0.55 and 1.6 μm are shown in Fig. 12. The retrieval

686 is then performed ignoring SRF. Fig. 13 shows the impact of SRF on the retrieval of SGS, SPS,

687 and SSA. For forward simulations without taking SRF into account (labeled as No in Fig. 13),

688 SGS, SPS, and SSA are well received as expected. And it agrees with Fig. 6. However, ignoring

689 the impact of SRF introduces about 7% uncertainties in the simulated surface reflectance and

690 this causes about 5-7% error in both SGS (overestimation) and SSA (underestimation). Taking

691 SRF into account leads to a smaller surface reflectance at 1.6 μm due to potential gas absorption

692 at this wavelength, thus introduces an overestimation for SGS. However, due to a significantly

693 smaller impact at 0.55 μm, the SRF does not play a significant role in the retrieval of SPS.

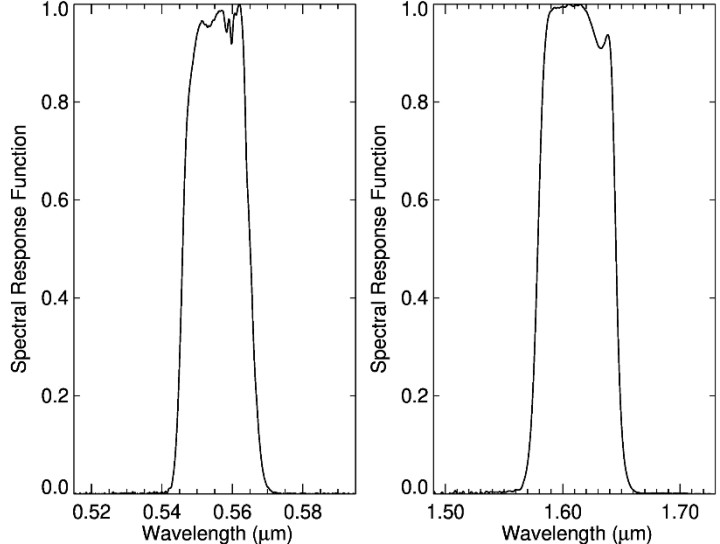

695 Fig. 12 Spectral response function of 0.55 (left) and 1.6 (right) μm of the SLSTR
696              instrument

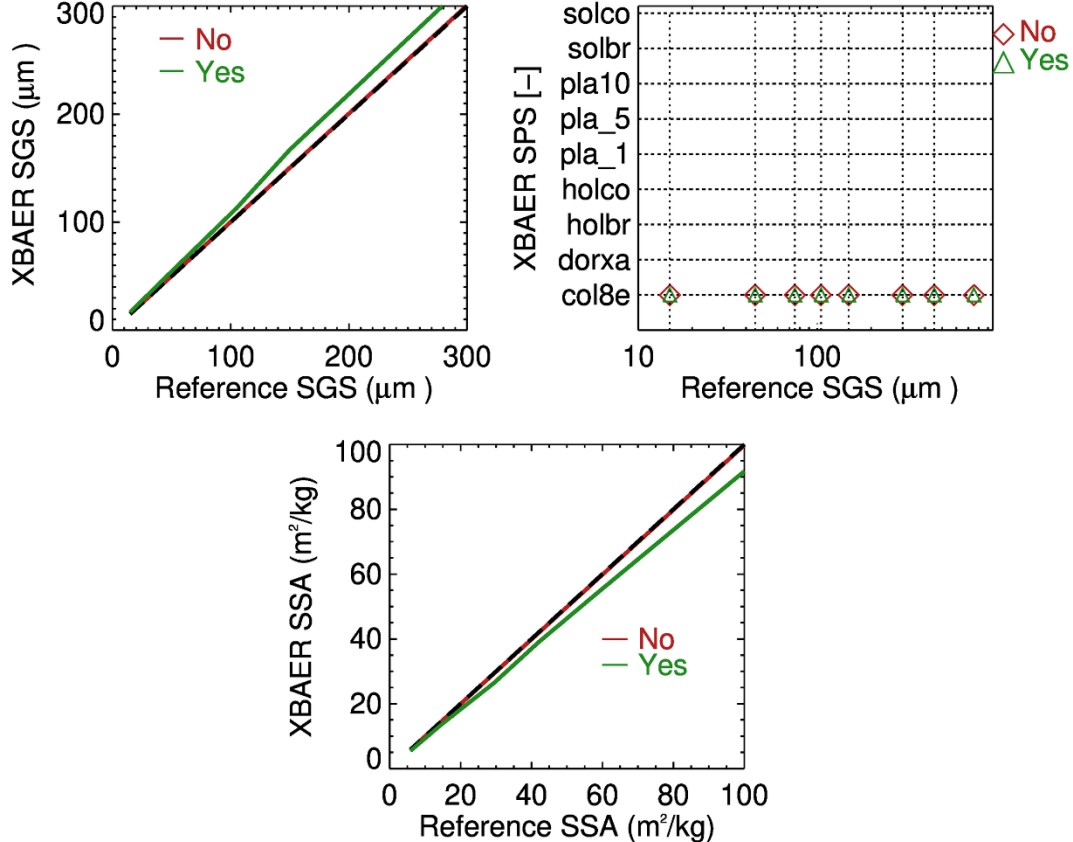

Fig. 13 Impact of SRFon the retrieval of SGS (upper left) SPS (upper right) and SSA (lower). Different colors indicate retrieval results without (No) and with (Yes) SRF in forward simulations, black dash line is the 1:1 line.

## 7.2 Impact of snow inhomogeneities

In this section, a realistic model of snow layer is represented by vertically inhomogeneous, polydisperse ice crystals habit mixture. Following Saito et al (2019), the gamma distribution with respect to the maximal dimension will be used to describe polydisperse properties:

$$n(D) = NG(D), \tag{9}$$

Here, N is the number of ice particles per unit volume, G(D) is the gamma distribution function, i.e.,

$$G(D) = C(D/v)^{k-1}e^{-D/v}, \tag{10}$$

where k and v are the shape and scale parameters, normalization factor C is defined as

$$C = \left[ \int\limits_{D_{\min}}^{D_{\max}} (D/v)^{k-1} e^{-D/v} dD \right]^{-1},$$
(11)

$D_{\min}$ and $D_{\max}$ describe the minimal and maximum particle sizes in the distribution.
In order to introduce the vertical inhomogeneity, we use the measurement of snow density and
equivalent optical diameter vertical profiles conducted during the SnowEx17 campaign.
Accounting for that the equivalent optical diameter cannot be directly used to define parameters
of Gamma distribution, we use the vertical profile as a shape of the mode (most frequent value
in a dataset), i.e.
$$D_0(z) = \frac{D_e(z)}{D_e(z_{top})} D_0(z_{top}),$$
(12)

where $D_e(z)$ is the measured vertical profile of equivalent optical diameter, $D_0(z)$ is the vertical
profile of the mode. The mode near the top of snow layer, $D_0(z_{top})$, we assume to be equal 400
μm according to the measurement data reproduced by Saito et al (2019) in Fig. A1.
Taking into account the analytical expression of the mode via shape and scale parameters,
$$D_0 = (k-1)v.$$
(13)

and the following relationship between shape and scale parameters derived by Saito et al (2019):
$$k = 11.38 v^{-0.167} - 2.$$
(14)

we can estimate parameters k and v of Gamma distribution corrsponding to $D_0(z)$ given by Eq

(12).

Snow Grain Habit Mixture (SGHM) model is used according to Saito et al (2019). In
particular, the particle habits include droxtal, solid hexagonal column, and solid bullet rosette.
Habit fraction, $f_h(D)$, as a function of maximal dimension of the SGHM model is presented in
the right panel of Fig. 15. The habit fraction is defined so that ,for each D,
$$\sum_{h=1}^{3} f_h(D) = 1.$$
(15)

The selected SGHM model enables us to derive the total volume of ice per unit volume of
air as

$$V_t = N \sum_{h=1}^{3} \left[ \int_{D_{\min}}^{D_{\max}} V_h(D) f_h(D) G(D) dD \right], \tag{16}$$

and ice water content (IWC)

$$IWC = V_t \rho_{ice}, \tag{17}$$

where $V_h(D)$ is the volume of each habit as given in database of Yang and $\rho_{ice}$ is the density of
ice.
Taking into account that the vettical profile if IWC is measured (see right panel of Fig.
14), we can obtain the vertical profile of particle number density. Using Eqs (16) and (17), we
have

$$N(z) = \frac{IWC(z)}{\rho_{ice} \sum_{h=1}^{3} \left[ \int_{D_{\min}}^{D_{\max}} V_h(D) f_h(D) G(D,z) dD \right]}. \tag{18}$$

Summing up, we define the microphysical properteis of snow layer using the following
model of particle size distribution

$$n(D,z) = N(z) C \left[ \frac{(\bar{k}-1)D}{D_0(z)} \right]^{\bar{k}-1} \exp\left[ -\frac{(\bar{k}-1)D}{D_0(z)} \right], \tag{19}$$

where $D_0(z)$ and $N(z)$ are given by Eq (12) and (18), respectively, shape parameter, $k$, is
assumed to be altitude independent and set to 2.3.
The bulk single-scattering properteis of snow layer such as extinction coeffiicent,
scattering coefficient and scattering function are defined by the same way as proposed by Baum
et al. (2011). For instance, the bulk extinction coefficient is calculated as

$$\beta_{ext}(z) = \int_{D_{\min}}^{D_{\max}} \left[ \sum_{h=1}^{3} \sigma_{ext,h}(D) f_h(D) n(D,z) dD \right], \tag{20}$$

where $\sigma_{ext,h}(D)$ is the extinction cross-section as given for each habit in database of Yang et
al.

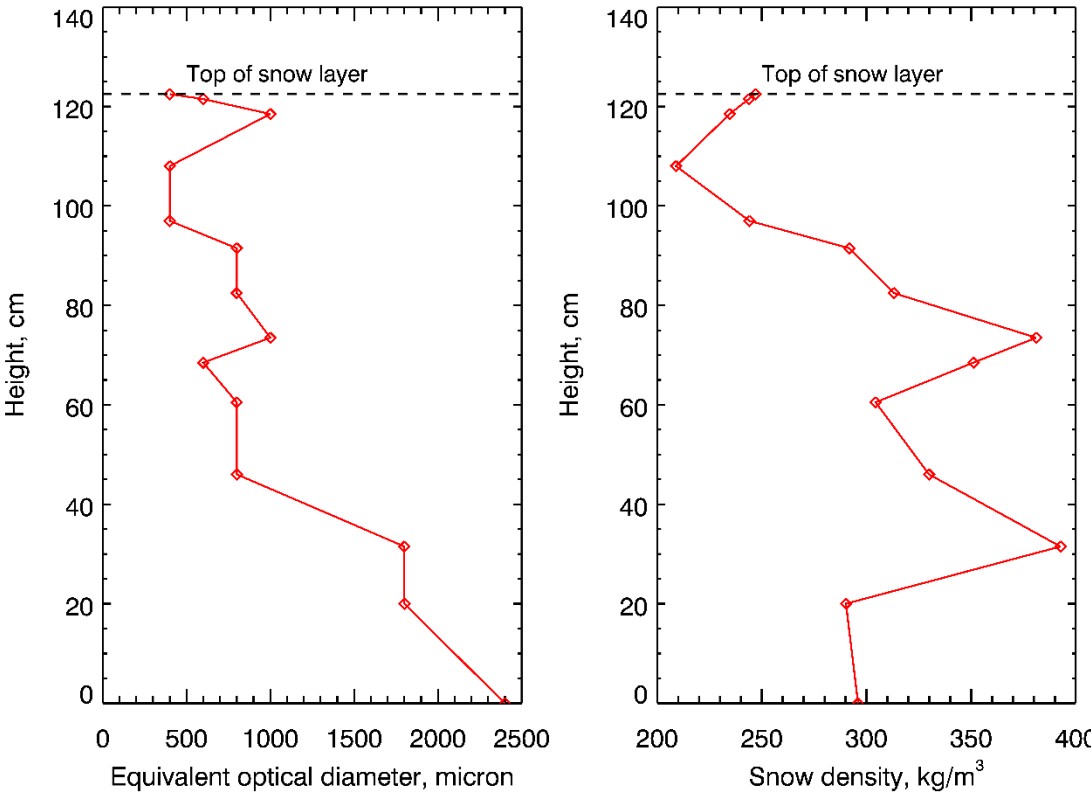


Fig. 14 Snow properties used for simulations to investigate the impacts of snow layer
model on XBAER retrieval (left) snow grain size profile and (right) snow density observed
during SnowEx17 campaign

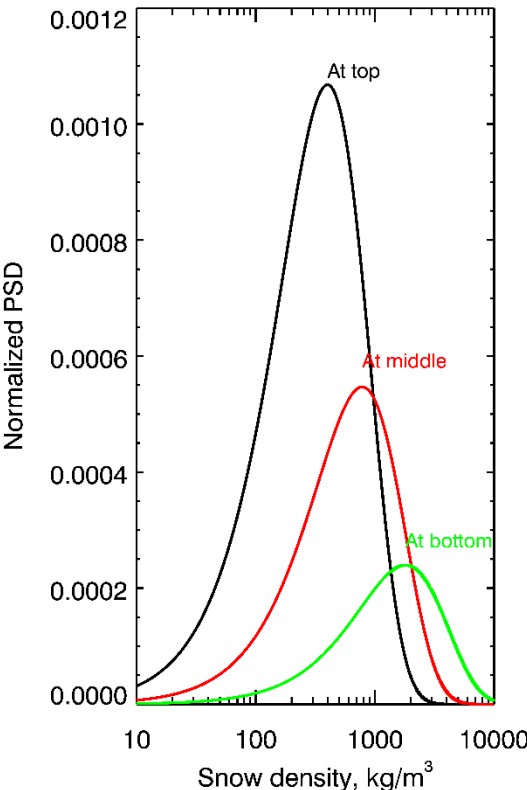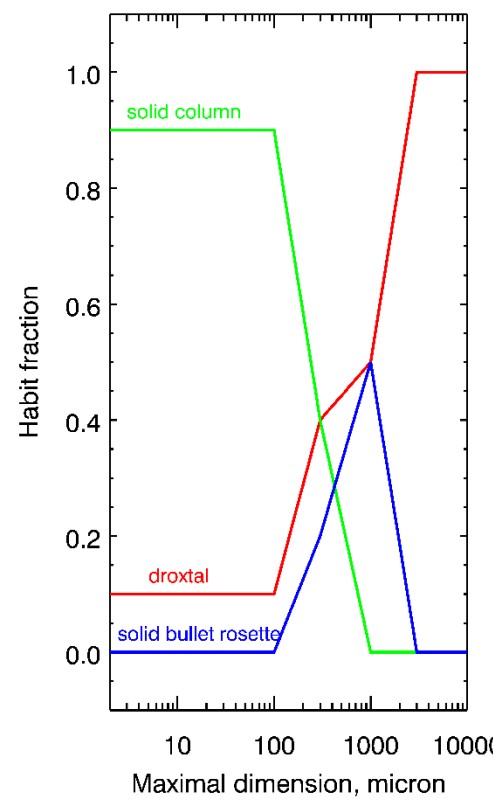


Fig. 15 Snow properties used for simulations to investigate the impacts of habit mixture

model on XBAER retrieval: (left) particle size distribution of snow grain size in snow layer;
(right) habit fraction suggested by Saito et al (2019)

The following settings are used to simulate the reference snow reflectance at wavelengths

0.55 and 1.6 μm;
➢ **Snow Layer:** vertically inhomogeneous, polydisperse habit mixture and model as
described above;
➢ **Atmosphere:** excluded.

Using the simulated reflectances in XBAER algorithm, we have retrieved SPS as dorxtal

with the maximal dimension equal to 740 μm. Taking into account that the model of PSD near
the top of snow layer is 400 μm and the mean value calculated as $kD_0/(k-1)$ is equal to 708 μm,
one can see that the retrieved maximal dimension is an estimation of mean value of PSD near
the top of snow layer.
Since there is no single reference SGS values when a PSD is used, it is important to check
the representativeness of XBAER derived SGS. Accounting for that the mode and mean values
for a given PSD are two typical "effective" way to describe polydisperse medium, we compared
reflectances of snow layer calculated assuming PSD in the form of Gamma distribution and
assuming monodisperse medium with SGS equal to the mode or to the mean of selected PSD.
In order to simplify analysis, we consider vertically homogeneous snow layer consisting of only
single particle habit. The calculations of reflectance were performed for severely roughened
aggregate of 8 columns and droxtal particles setting the shape parameter, k, equal to 2.3 and
the model equal to [100, 300, 500, 700, 1000, 2000, 3000, 5000] µm.
Fig. 16 shows the comparison of snow reflectance calculated assuming monodisperse and
polydisperse snow model. In the case of monodisperse model, SGS is assumed to be equal to
the mean or to the mode value of PSD. We can see that the surface reflectance calculated using
the mean value of PSD agrees better with reference values, tan reflectance calculated using the
mode value. In particular, the root-mean-square deviation (RMSE) values are more than 2 times
smaller. One can also see from Fig. 16 that the difference between monodisperse reflectances
calculated using mean or mode PSD values decreases with increase of the PSD mode. It can be
explained due to the fact that the increase of PSD mode leads to the increase of absorption and
decrease of reflectance sensitivity with respect to the variation of SGS.

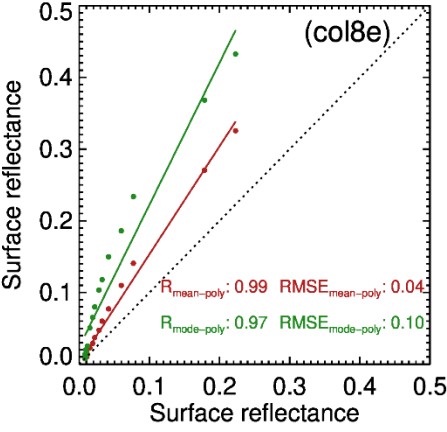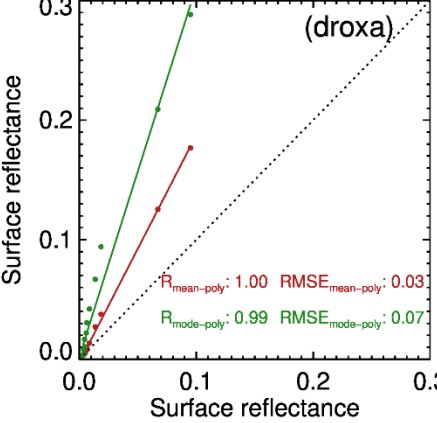


Fig. 16 The comparison between simulated snow reflectance using mono and poly-disperse
snow model consisting of aggregate of columns (left) and droxtal (right). In the case of
monodisperse model, the SGS is assumed to be the mean and mode value of PSD at the top of
snow layer (see left panel of Fig. 14). The reference value is shown on the x-axis.

## 8 Conclusions

SGS, SPS, and SSA are three important parameters to describe snow properties. They play
important roles in the changes in snow albedo/reflectance and impact the atmospheric and
energy-exchange processes. A better knowledge of SGS, SPS, and SSA can provide more
accurate information to describe the impact of snow on Arctic amplification processes. The
information about SGS, SPS, and SSA may also explore new applications to understand the
atmospheric conditions (e.g. aerosol loading). Although some previous attempts (e.g. Lyapustin
et al., 2009) show the capabilities of using passive remote sensing to derive SGS over a large
scale, no publication has been found to derive SGS, SPS, and SSA simultaneously. To our best
knowledge, this is the first paper, attempting to retrieve these parameters using satellite
observations.
The new algorithm is designed within the framework of the XBAER algorithm. The
XBAER algorithm has been applied to derive SGS, SPS, and SSA using the newly launched
SLSTR instrument onboard Sentinel-3 satellite. This is the first part of the paper, to describe
the algorithm, and to present the sensitivity studies.
The SGS, SPS, and SSA retrieval algorithm is based on the recent publication by Yang et
al., (2013), in which a database of optical properties for nine typical SPSs (aggregate of 8
columns, droxtal, hollow bullet rosettes, hollow column, plate, aggregate of 5 plates, aggregate
of 10 plates, solid bullet rosettes, column) are provided. Previous publications show that this
database can be used to retrieve ice crystal properties in both ice cloud and snow (e.g., Järvinen
et al., 2018; Saito et al., 2019). The algorithm is a LUT-based approach, in which the
minimization is achieved by the comparison between atmospherically corrected TOA
reflectance at 0.55 and 1.6 μm observed by SLSTR and pre-calculated LUT of surface
reflectances under different geometries and snow properties. The retrieval is relatively time-
consuming because the minimization has to be performed for each SPS and the optimal SGS
and SPS are selected after 9 minimizations are done. The SSA is then obtained using the
retrieved SGS and SPS based on another pre-calculated LUT.
The sensitivity studies with respect to the impacts of SPS, ICSR, aerosol and cloud
contamination on XBAER derived SGS and SPS provide a comprehensive understanding of
the retrieval accuracy of the new algorithm. The main findings of the theoretical considerations
are: (1) XBAER derived SGS is more likely to represent the average SGS near the top of snow
layer when a PSD is known; (2) SPS plays an important role for the retrieval accuracy of SGS,
the retrieved SGS can differ several times by usage different SPSs in the retrieval process; (3)
Impact of ICSR on the retrieval accuracy of SGS can be neglected, ignoring ICSR completely
may introduce maximal 3% error on the retrieval accuracy of SGS, especially for large ice
crystals; (4) Assumption of convex particle shape (e.g., sphere) of a non-convex ice crystal
leads to the underestimation of the retrieved SSA; (5) The impact of aerosol and cloud increase
with the increase of both aerosol/cloud loading and SGS; (6) The impact of instrument SRF
may introduce some positive biase for SGS and negative bias for SSA, however, it plays no
role for the determination of SPS
Even though all major possible factors affecting the retrieval accuracy of XBAER
algorithm are investigated in this paper, in reality, the final retrieval accuracy can only be
evaluated by performing a thorough comparison with independent measurement results because
uncertainties caused by each individual factor can compensate each other in the real satellite
retrieval. All details of such validation can be found in the companion paper of Mei et al (2020c).

**Code and data availability**
The code and used dataset can be found at iup.uni-bremen.de/sciatran/

**Author contributions**
LM and VR conceptualized the study, LM, VR and CP implemented the code and processed
the data. LM and VR analyzed the data. LM and VR prepared he manuscript with contribution
from all co-authors. LM, VR and JB polished the whole manuscript.

**Competing interests**

The authors declare that they have no conflict of interest.

**Acknowledgements**

This research was funded by the Deutsche Forschungsgemeinschaft (DFG, German Research

Foundation) – Project-ID 268020496 – TRR 172. The SLSTR data is provided by ESA. We

thank the valuable discussion with Dr. M Saito.

**Appendix**

According to the definition of specific surface area

$$SSA = \frac{A}{\rho V},\qquad\qquad(A1)$$

one needs to calculate the total area A of ice crystal. In the following sections, we consider

in details the basic equations to calculate total area and SSA of different SPSs given in

database of Yang et al (2013) and used above within the retrieval algorithm.

➢ **Droxtal, solid column, plate**

In the case of convex faceted particles such as droxtal, solid column, and plate, the

calculation of total area is straightforward and based on the Cauchy's surface area formula:

$$A = 4A_p.\qquad\qquad(A2)$$

Taking into account that for selected SPS, one can find corresponding V and $A_P$ in database

given by Yang et al., (2013), we have the following results for SSA of such particles:

$$SSA = \frac{4A_p}{\rho V}.$$
(A3)

➢ **Hollow column**
In this case a solid column includes two equal cavities in the form of a hexagonal
pyramid and cannot be considered as convex particle. The aspect ratio of hollow column
with the height, d, of hexagonal pyramid is given according to Yang et al., (2013) as:
$$\frac{2a}{L} = \begin{cases} 0.7, & L < 100\,\mu m \\ \dfrac{6.96}{\sqrt{L}}, & L \geq 100\,\mu m \end{cases}, \quad d = 0.25L.$$
(A4)

The volume of such hollow column is given by
$$V = V_c - 2V_p,$$
(A5)

where the volume of solid column, $V_c$, and a hexagonal pyramid, $V_p$, are,
$$V_c = \frac{3\sqrt{3}}{2} a^2 L,$$
(A6)

$$V_p = \frac{\sqrt{3}}{2} a^2 d.$$
(A7)

Thus, the volume, V, is
$$V = \frac{\sqrt{3}}{2} a^2 (3L - 2d).$$
(A8)

Employing the relationship between d and L given by Eq (A4) and excluding a, we
have
$$V = \frac{2.5\sqrt{3}}{2} a^2 L = \begin{cases} m_0 m_1^2 L^3, & L < 100\,\mu m \\ m_0 m_2^2 L^2, & L \geq 100\,\mu m \end{cases},$$
(A9)

where $m_0 = 2.5 \times \sqrt{3}/2$, $m_1 = \dfrac{0.7}{2}$, and $m_2 = \dfrac{6.96}{2}$. For a selected volume, $V$, the
length, $L$, is calculated as follows:
$$L = \begin{cases} [V/m_0/m_1^2]^{\frac{1}{3}}, & V < V_{100} \\ [V/m_0/m_2^2]^{\frac{1}{2}}, & V \geq V_{100} \end{cases}, \quad \text{(A10)}$$

where $V_{100} = m_0 m_2^2 100^2.$
Let us now calculate the area of each triangle side of the pyramid
$$S_t = \frac{a}{2}\sqrt{d^2 + \frac{3a^2}{4}}. \quad \text{(A11)}$$

The area of lateral surface of two pyramids is
$$S_p = 3a\sqrt{4d^2 + 3a^2}. \quad \text{(A12)}$$

And the total surface area of hollow column is given by
$$S = 6aL + 3a\sqrt{4d^2 + 3a^2}, \quad \text{(A13)}$$

where $a$ and $d$ should be expressed via $L$ according to Eq. (A4).
Having obtained the total area, one can calculate specific surface area
$$SSA = \frac{S}{\rho V}, \quad \text{(A14)}$$

➢ **Hollow bullet rosettes**
In this case a solid column includes a cavity in the form of a hexagonal pyramid with
height $H$ and a hexagonal pyramid with height $t$ on the opposite site of column. The aspect
ratio and parameters $H$ and $t$ is given according to Yang et al., (2013) as:
$$\frac{2a}{L} = 2.3104 L^{-0.37}, \quad t = \frac{\sqrt{3}a}{2\tan(28^\circ)}, \quad H = 0.5(t+L). \tag{A15}$$
The volume of a hollow bullet rosettes is given by
$$V_1 = V_c - V_- + V_+. \tag{A16}$$
Using Eqs. (A6) and (A7), we have
$$V_1 = \frac{3\sqrt{3}}{2}a^2 L - \frac{\sqrt{3}}{2}a^2 H + \frac{\sqrt{3}}{2}a^2 t = \frac{\sqrt{3}}{2}a^2(3L - H + t). \tag{A17}$$
Substituting H as given by Eq (A15), we obtain
$$V_1 = \frac{\sqrt{3}a^2}{4}(5L + t). \tag{A18}$$
Using Eq (A15), we express parameters $a$ and $t$ of hollow bullet rosettes via $L$:
$$a = m_a L^{p_a}, \tag{A19}$$
$$t = m_t m_a L^{p_a}, \tag{A20}$$
where coefficients, $m_a$, $m_t$, and $p_a$ are
$$m_a = \frac{2.3104}{2}, m_t = \frac{\sqrt{3}}{2\tan(28^\circ)}, p_a = 1 - 0.37. \tag{A21}$$
The expression (A18) can be rewritten as:
$$V_1 = \frac{\sqrt{3}}{4}m_a^2 L^{2p_a+1}(5 + m_t m_a L^{-0.37}). \tag{A22}$$
The total area of hollow bullet rosette is written as
$$S_1 = 6aL + \frac{3a}{2}\sqrt{4H^2 + 3a^2} + \frac{3a}{2}\sqrt{4t^2 + 3a^2}$$ (A23)
and can be calculated when for a selected maximal dimension D the parameter L is
known. For a desired dimension D (volume V) of hollow bullet rosettes, consisting of
6 equal rosettes (See Table 1), Eq (A22) was solved with respect to the length, L, using
following iterative approach:
$$L_n = \left[ \frac{2V}{3\sqrt{3}m_a^2(5 + m_t m_a L_{n-1}^{-0.37})} \right]^{\frac{1}{2p_a + 1}}.$$ (A24)
The iterative process starts with $L_0 = 1$ and finishes when $\left| \frac{L_n - L_{n-1}}{L_n} \right| \leq 10^{-4}$. The total
area of hollow bullet rosettes is calculated as;
$$S_1 = 6aL + \frac{3a}{2}\sqrt{4H^2 + 3a^2} + \frac{3a}{2}\sqrt{4t^2 + 3a^2}.$$ (A25)
The SSA is given by
$$SSA = \frac{6S_1}{\rho V}.$$ (A26)
➢ **Solid bullet rosettes**
The aspect ratio and parameter *t* are given according to Yang et al., (2013) as:
$$\frac{2a}{L} = 2.3104 L^{-0.37}, \ t = \frac{\sqrt{3}a}{2\tan(28^\circ)}.$$ (A27)
The volume of single solid bullet rosette is
$$V_1 = V_c + V_+.$$ (A28)
Using Eq. (A6), we have
$$V_1 = \frac{3\sqrt{3}}{2}a^2L + \frac{\sqrt{3}}{2}a^2t = \frac{\sqrt{3}}{2}a^2(3L+t).$$   (A29)
Using formula given by Eq (A27), we express parameters $a$ and $t$ of solid bullet rosette
via $L$:
$$a = m_a L^{p_a},$$   (A30)
$$t = m_t m_a L^{p_a},$$   (A31)
Where coefficients, $m_a$, $m_t$, and $p_a$ are the same as in the case of hollow bullet
rosette given by Eq. (A21). The expression (A29) can be rewritten as
$$V_1 = \frac{\sqrt{3}}{2}m_a^2 L^{2p_a+1}(3 + m_t m_a L^{-0.37}).$$   (A32)
For a desired volume $V$ of solid bullet rosettes, consisting of 6 equal rosettes (see
Table 1), this equation was solved with respect to the length, $L$, of the solid bullet rosette
using following iterative approach:
$$L_n = \left[ \frac{V}{3\sqrt{3}m_a^2(3 + m_t m_a L_{n-1}^{-0.37})} \right]^{\frac{1}{2p_a+1}}.$$   (A33)
The total area of solid bullet rosettes is calculated as;
$$S_1 = 6aL + \frac{3\sqrt{3}a^2}{2} + \frac{3a}{2}\sqrt{4t^2 + 3a^2}.$$   (A34)
The SSA is given by
$$SSA = \frac{6S_1}{\rho V}.$$   (A35)

➢ **Aggregate of 5 and 10 plates**

According to the paper of Yang et al (2013), Table 1 provides the aspect ratios of the

ice crystal habits. In the case of an aggregate of columns or plates, the semi-width $a$ and
length $L$ of each hexagonal element of the aggregate are on a relative scale. In order to covert
these parameters in absolute values, let us consider the following relationship given in Yang
et al (2013) for aspect ratio of plate:
$$\frac{2a}{L} = \begin{cases} 1, & a \le 2\,\mu m \\ m_1 a + m_0, & 2 < a < 5\,\mu m \\ m a^p, & a \ge 5\,\mu m \end{cases} \tag{A36}$$

where constants are: $m_1$=0.2914, $m_0$=0.4172, $m$=0.8038, $p$=0.526.

Using this expression and accounting for that relative values for a, given in Table 1, are

greater than 5μm, we can express $L_r$ via $a_r$ as
$$L_r = \frac{2a_r}{m a_r^p} = \frac{2 a_r^{1-p}}{m}. \tag{A37}$$

where subscript $r$ denotes that they are on relative scale. The volume of a hexagonal plate on
relative scale is given by
$$v_r = \frac{3\sqrt{3}}{2} a_r^2 L_r = \frac{3\sqrt{3}}{m} a_r^{3-p}. \tag{A38}$$

The volume of aggregates of 5 or 10 plates is given by
$$V_r = \frac{3\sqrt{3}}{m} \sum_{i=1}^{N} a_{r,i}^{3-p}, \tag{A39}$$

where $N$=5 and $N$=10 for 5 and 10 plates, respectively. The absolute value of the volume,

$V$, for a selected maximal dimension of aggregate of 5 or 10 plates one can find in database
presented by Yang et al (2013). Introducing the scaling factor
$$C = \frac{V_r}{V}, \tag{A40}$$

We rewrite expression (A38) as
$$V = \frac{V_r}{C} = \frac{3\sqrt{3}}{mC} \sum_{i=1}^{N} a_{r,i}^{3-p} = \frac{3\sqrt{3}}{m} \sum_{i=1}^{N} a_i^{3-p}, \tag{A41}$$

where the absolute value of semi-width, $a_i$, is given by
$$a_i = \frac{a_{r,i}}{C^{(3-p)^{-1}}},$$
(A42)

Having obtained the absolute value of $a_i$ for each plate, the length $L_i$ is calculated as:
$$L_i = \begin{cases} 2a_i, & a \le 2\,\mu m \\ \dfrac{2a_i}{m_1 a_i + m_0}, & 2 < a < 5\,\mu m \\ \dfrac{2a^{(1-p)}}{m}, & a \ge 5\,\mu m \end{cases}$$
(A43)


The total area of a hexagonal plate with semi-width $a_i$ and length $L_i$ is given by
$$S_i = 2\frac{3\sqrt{3}}{2}a_i^2 + 6a_i L_i,$$
(A44)

The total area is given by
$$S = \sum_{i=1}^{N} S_i.$$
(A45)

Having obtained the total area, one can calculate SSA as the total surface area of a material per
unit of mass:
$$SSA = \frac{S}{\rho V},$$
(A46)

where ρ=917 kg/m$^3$ is the density of ice.

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
