# Peer review of "The retrieval of snow properties from SLSTR/"

_The Cryosphere, 2020_

## Referee Comment (RC1) · Anonymous Referee #1 · 2 Nov 2020

The paper describes a comprehensive sensitive study of a new retrieval algorithm to derive snow properties using passive remote sensing. The topic fits TC. The paper is well-written and easy to follow. The main findings in this paper are important for the whole community. For instance, the impact of the ice crystal shape and atmospheric effects on the snow properties retrieval will provide very valuable information for the whole community. Moreover, the authors try to explain the understanding gap between the remote sensing community and the campaign-based community and made a great effort to minimize such gaps. I would suggest the paper to be published after addressing the following minor aspects.

[Figure]

The authors highlight that the current snow shape assumption, such as spherical shape, may be valid for snow albedo estimation, but not for the directional reflectance estimation, due to the different impacts of ice crystal shape on the phase function and extinction/absorption coefficient, by citing the previous publication, it will be helpful if the author can extend the explanation of this part (L134), does the snow albedo purely depend on the particle extinction/absorption coefficient?

I don't think that the SSA is an independent retrieved parameter in the XBAER algorithm, although the author has mentioned in the abstract and also in the introduction part, it will provide a better understanding for the reader, that is, are the snow size and shape the two fundamental inputs for the XBAER algorithm, especially, as the author highlighted, the ice crystal shape cannot be precisely validated and I think it is also very difficult for the user to use the shape rather than SSA, so can the author directly retrieve size and SSA?

Some names, for instance, snow particle shape, ice crystal shape should be harmonized in the paper

Why there are obvious oscillations in Fig 2 and Fig. 3, especially for the phase function?

What is the typical valid range for SSA? How strong the SSA variability is?

What is the physical reason behind that the roughness plays such a minor role in the snow properties retrieval? Are those definitions of the snow surface roughness, with those three values, reasonable? Or those values themselves are too small?

The authors make a good effort to investigate especially the aerosol impact, I can clearly see that aerosol play a very important role in the retrieval, the authors propose some Arctic aerosol scenarios, however, snow occurs also in high polluted regions, at least during winter, the authors should include some explanation for this situation.

---

## Referee Comment (RC2) · Anonymous Referee #2 · 21 Dec 2020

Review of manuscript submitted to The Cryosphere:

Title: The retrieval of snow properties from SLSTR/ Sentinel-3 - part 1: method description and sensitivity study

Authors: Linlu Mei et al.

**General Comments**

The authors discuss a method to retrieve three snow properties: (i) snow grain size (SGS), (ii) snow particle shape (SPS), and (iii) specific surface area (SSA) from data collected with a two-channel radiometer with dual-viewing observation capabilities flown on the Sentinel-3 satellite. Supposedly this article is just the first part of a two-part paper describing the retrieval algorithm (with acronym XBAER). Although the authors claim in the abstract that XBAER "has been applied on the Top-Of-Atmosphere reflectance measured by the Sea and Land Surface Temperature Radiometer (SLSTR) instrument onboard Sentinel-3 to derive snow properties", no evidence of any such application is provided in the present article. Hence, there is no way of judging from the work presented in this article how this method will perform when applied to actual data collected with the SLSTR instrument. I therefore recommend that the authors merge the essence of this paper with part II that I expect will present results of applications of the retrieval algorithm to data collected with the SLSTR instrument.

The authors provide a useful discussion of how a database of optical properties of ice particles developed by Ping Yang and co-authors for application to cirrus clouds can be used also to study snow properties. Their study of the impact of model parameter uncertainties in Section 5 is also useful. But the paper has some important limitations:

1. Any credible remote sensing algorithm has to include a robust cloud screening tool. How to construct such a cloud screening tool for the SLSTR instrument is a challenge that warrants serious consideration. As alluded to in Section 6 of the paper, identifying and removing the contribution of an ice cloud to the measured signal is a very challenging problem as discussed by Chen et al. (2104, 2018).

2. Atmospheric correction is another important issue that requires serious consideration. A revised version of this paper merged with the promised part II must include a discussion of this issue.

3. The description of the SLSTR instrument in Section 2.1 lacks important information about the spectral response of the two channels at 0.55 $\mu$m and 1.6 $\mu$m. As discussed by Chen et al. (2017) the response function of the 1.6 $\mu$m channel requires careful consideration because the optical properties of snow/ice varies considerably over small wavelength intervals in the 1.6 $\mu$m spectral range.

4. The SLSTR instrument has a total of nine channels. The authors provide no explanation for why they have decided to use only two of those channels for this work. It would seem that using more channels should be helpful.

5. In Section 2.1 the authors state "The statistical analysis has been performed using observations over Greenland during April and September 2017. April and September are reported to be representativeness months of the Arctic..." This focus on the months of April and September only is a serious limitation. We would like to know how the snow properties evolve over the summer season from the beginning of the melt in April/May to the freeze-up in August/September.

6. In Section 4 describing the XBAER algorithm, the authors state "The first stage includes the estimation of SGS using the effective Lambertian surface albedo after atmospheric correction ...". The authors must explain what is meant by "effective Lambertian surface albedo" and also how "atmospheric correction" is performed. This information is essential.

7. The modeling carried out in this paper is based on the assumption that the snowpack is vertically homogeneous and consisting of a mono-dispersion of snow particles of a pre-defined shape. In reality these assumptions are not fulfilled. Also, the light penetration depth at wavelength 1.6 $\mu$m is much shorter than at 0.55 $\mu$m. The impact of these circumstances are not discussed.

**Specific Comments**

- The Bidirectional Reflectance Factor (BRF) is introduced at line 206, but defined only at line 307.

- The authors state (line 445) that field-measurements of SSA are based on the assumption that snow grains have spherical shapes. What are the uncertainties in the field-measured SSA values incurred by this assumption?

**References**

Chen, N., W. Li, T. Tanikawa, M. Hori, T. Aoki, and K. Stamnes, Cloud mask over snow/ice covered areas for the GCOM-C1/SGLI cryosphere mission: Validations over Greenland, J. Geophys. Res. Atmos., 119, 12,287-12,300, 2014. doi: 10.1002/2014JD022017.

Chen, N., W. Li, T. Tanikawa, M. Hori, R. Shimada, Te. Aoki, and K. Stamnes, Fast yet accurate computation of radiances in shortwave infrared satellite remote sensing channels, Opt. Express, 17, 649-664, 2017.

Chen N., W. Li, C. Gatebe, T. Tanikawa, M. Hori, R. Shimada; T. Aoki, and K. Stamnes, New cloud mask algorithm based on machine learning methods and radiative transfer simulations, Remote Sensing of the Environment, 219, 62-71, 2018.

---

## Author Comment (AC1) · 5 Feb 2021

Dear Editor, dear reviewer,

Thanks for the valuable comments, which help to improve the quality of the paper. The detailed replies are addressed below point by point in blue.

Best regards,

Linlu Mei on behalf of all co-authors

The paper describes a comprehensive sensitive study of a new retrieval algorithm to derive snow properties using passive remote sensing. The topic fits TC. The paper is well-written and easy to follow. The main findings in this paper are important for the whole community. For instance, the impact of the ice crystal shape and atmospheric effects on the snow properties retrieval will provide very valuable information for the whole community. Moreover, the authors try to explain the understanding gap between the remote sensing community and the campaign-based community and made a great effort to minimize such gaps. I would suggest the paper to be published after addressing the following minor aspects.

The authors highlight that the current snow shape assumption, such as spherical shape, may be valid for snow albedo estimation, but not for the directional reflectance estimation, due to the different impacts of ice crystal shape on the phase function and extinction/absorption coefficient, by citing the previous publication, it will be helpful if the author can extend the explanation of this part (L134), does the snow albedo purely depend on the particle extinction/absorption coefficient?

Response: As presented in the cited paper such as Jin et al (2008), directional quantities, such as bidirectional reflectance and radiance, are however more sensitive to scattering phase function and hence to particle shape, while the hemispherically averaged radiative quantities, such as albedo is not very sensitive to the finer aspects of the scattering phase function.

The snow albedo does not purely depend on particle extinction/absorption coefficient, but also on how those single particle aggregates.

We included more explanations in the revised version.

Jin, Z., Charlock, T. P., Yang, P., Xie, Y., & Miller, W. (2008). Snow optical properties for different particle shapes with application to snow grain size retrieval and MODIS/CERES radiance comparison over Antarctica. Remote Sensing of Environment, 112(9), 3563–3581. doi:10.1016/j.rse.2008.04.011

I don't think that the SSA is an independent retrieved parameter in the XBAER algorithm, although the author has mentioned in the abstract and also in the introduction part, it will provide a better understanding for the reader, that is, are the snow size and shape the two fundamental inputs for the XBAER algorithm, especially, as the author highlighted, the ice crystal shape cannot be precisely validated and I think it is also very difficult for the user to use the shape rather than SSA, so can the author directly retrieve size and SSA?

Response: There are at least two different manners to describe the snow properties, one is to use the combination of ice crystal shape and grain size as inputs, the other is to assume that the snowpack is a medium consisting of grains and bubbles. However, in both way, there are parameters, which cannot be precisely validated in the reality. For the size-shape manner, the particle shape is difficult to be validated while the mean photo path length cannot be evaluated for the grains-bubbles way. So it is impossible to estimate SSA without certain assumption in advance. In our case, we cannot retrieve SSA without a knowing or assuming ice particle shape.

Some names, for instance, snow particle shape, ice crystal shape should be harmonized in the paper

Response: The names are harmonized in the revised version.

Why there are obvious oscillations in Fig 2 and Fig. 3, especially for the phase function?

Response: The oscillations comes from the original Yang's database.

What is the typical valid range for SSA? How strong the SSA variability is?

Response: The valid range of SSA depends on the snow properties, and it differs from region to region. For instance, in the paper of Picard et al (2009), the SSA varies from $0 - 35$ m$^2$/kg while the results from Kokhanovsky et al (2019) shows a range of $0 - 80$ m$^2$/kg. And the variabilities of SSA is quite large, depending on the snow metamorphism due to changes in thermodynamic conditions.

Picard, G., Arnaud, L., Domine, F., & Fily, M. (2009). Determining snow specific surface area from near-infrared reflectance measurements: Numerical study of the influence of grain shape. Cold Regions Science and Technology, 56(1), 10–17. doi:10.1016/j.coldregions.2008.10.001

Kokhanovsky, A.; Lamare, M.; Danne, O.; Brockmann, C.; Dumont, M.; Picard, G.; Arnaud, L.; Favier, V.; Jourdain, B.; Le Meur, E.; Di Mauro, B.; Aoki, T.; Niwano, M.; Rozanov, V.; Korkin, S.; Kipfstuhl, S.; Freitag, J.; Hoerhold, M.; Zuhr, A.; Vladimirova, D.; Faber, A.-K.; Steen-Larsen, H.C.; Wahl, S.; Andersen, J.K.; Vandecrux, B.; van As, D.; Mankoff, K.D.; Kern, M.; Zege, E.; Box, J.E. Retrieval of Snow Properties from the Sentinel-3 Ocean and Land Colour Instrument. *Remote*

*Sens.* 2019, *11*, 2280. https://doi.org/10.3390/rs11192280

What is the physical reason behind that the roughness plays such a minor role in the snow properties retrieval? Are those definitions of the snow surface roughness, with those three values, reasonable? Or those values themselves are too small?

Response: The roughness defined in the Yang database describes the roughness of each ice crystal particle, not the roughness of the snow layer (or the surface homogeneity). And those three values (and we believe they represent the typical snow conditions) are provided by the Yang database, thus no other values can be used for the test. Please be noted, that the surface roughness of ice crystal may occurs for ice cloud, but it is quite rare that it may occurs in case of snow on the ground due to much slower and small surface irregularities, which are bound to disappear very fast because of basic thermodynamics (Colbeck, 1980, 1983). However, we would prefer to perform a comprehensive sensitive study, including all possible ice crystal properties into account, thus, we believe that the test of roughness is still needed and useful. A similar investigation of impact of snow particle roughness is also presented in Picard et al (2009).

Colbeck, S. C.: Thermodynamics of snow metamorphism due to variations in curvature, J. Glaciol., 26, 291-301, 10.3189/S0022143000010832, 1980.

Colbeck, S. C.: Theory of metamorphism of dry snow, J. Geophys. Res., 88, 5475-5482, 1983.

Picard, G., Arnaud, L., Domine, F., & Fily, M. (2009). Determining snow specific surface area from near-infrared reflectance measurements: Numerical study of the influence of grain shape. Cold Regions Science and Technology, 56(1), 10–17. doi:10.1016/j.coldregions.2008.10.001

The authors make a good effort to investigate especially the aerosol impact, I can clearly see that aerosol play a very important role in the retrieval, the authors propose some Arctic aerosol scenarios, however, snow occurs also in high polluted regions, at least during winter, the authors should include some explanation for this situation.

Response: The impact of aerosol, as we presented in the paper, is one of the most important parameters, needing to be addressed. For regions with strong pollution condition, the uncertainties of the retrieval will be larger. This condition may be quite critical over relatively lower latitude regions, where pollution may transport to snow covered regions. We added some more explanation in this section in the revised version.

---

## Author Comment (AC2) · 5 Feb 2021

Dear Editor, dear reviewer,

Thanks for the valuable comments, which help to improve the quality of the paper. The detailed replies are included below point by point in blue. We believe that all comments raised by the reviewer are addressed in the revised version. Please be aware that part 2 is not under preparation, it is already submitted together with this part 1 as companion papers.

Best regards,

Linlu Mei on behalf of all co-authors

**General Comments**

The authors discuss a method to retrieve three snow properties: (i) snow grain size (SGS), (ii) snow particle shape (SPS), and (iii) specific surface area (SSA) from data collected with a two- channel radiometer with dual-viewing observation capabilities flown on the Sentinel-3 satellite. Supposedly this article is just the first part of a two-part paper describing the retrieval algorithm (with acronym XBAER). Although the authors claim in the abstract that XBAER "has been applied on the Top-Of-Atmosphere reflectance measured by the Sea and Land Surface Temperature Radiometer (SLSTR) instrument onboard Sentinel-3 to derive snow properties", no evidence of any such application is provided in the present article. Hence, there is no way of judging from the work presented in this article how this method will perform when applied to actual data collected with the SLSTR instrument. I therefore recommend that the authors merge the essence of this paper with part II that I expect will present results of applications of the retrieval algorithm to data collected with the SLSTR instrument.

Response: We have clearly defined the usage of SLSTR in both Part 1 and Part 2. And we believe the part of "has been applied on the Top-Of-Atmosphere reflectance measured by the Sea and Land Surface Temperature Radiometer (SLSTR) instrument onboard Sentinel-3 to derive snow properties " is more proper to be presented like "has been tested on the observation characteristic of the Sea and Land Surface Temperature Radiometer (SLSTR) instrument onboard Sentinel-3", which is also true because all our theoretical investigations (part 1) are performed not on some random observation geometries, but on geometries obtained based on a huge statistical analysis of the real SLSTR observations, as presented in Fig. 1 in the paper.

We appreciate that the comments raising by the reviewer here, however, taking the content of each part and the length of papers into account, we believe that keep these two parts separated is more reasonable.

At the meantime, we decide to move some very complicated but important content from appendix to the main paper in part 1, which will again enhance the "difference" between the theoretical part 1 and practical part 2.

And we are extending the part 2 by more validation worldwide, the validation will be extended from just 1 month to more than 10 years in total. This will again create more trouble for reader to have a merged part 1 and part 2.

The authors provide a useful discussion of how a database of optical properties of ice particles developed by Ping Yang and co-authors for application to cirrus clouds can be used also to study snow properties. Their study of the impact of model parameter uncertainties in Section 5 is also useful. But the paper has some important limitations:

1.    Any credible remote sensing algorithm has to include a robust cloud screening tool. How to construct such a cloud screening tool for the SLSTR instrument is a challenge that warrants serious consideration. As alluded to in Section 6 of the paper, identifying and removing the contribution of an ice cloud to the measured signal is a very challenging problem as discussed by Chen et al. (2104, 2018).

Response: We agree with what the reviewer point out, with respect to the challenges of cloud screening over snow. However, with more than 10 years' experience of usage AATSR/SLSTR data in our group, with the continues developments and several publications for this topic, we believe, our cloud screening algorithm can provide good separation of cloud from snow.

Istomina, L. G., von Hoyningen-Huene, W., Kokhanovsky, A. A., and Burrows, J. P.: The detection of cloud-free snow-covered areas using AATSR measurements, Atmos. Meas. Tech., 3, 1005–1017, https://doi.org/10.5194/amt-3-1005-2010, 2010.

Jafariserajehlou, S., Mei, L., Vountas, M., Rozanov, V., Burrows, J. P., and Hollmann, R.: A cloud identification algorithm over the Arctic for use with AATSR–SLSTR measurements, Atmos. Meas. Tech., 12, 1059–1076, https://doi.org/10.5194/amt-12-1059-2019, 2019.

Istomina, L., Marks, H., Huntemann, M., Heygster, G., and Spreen, G.: Improved cloud detection over sea ice and snow during Arctic summer using MERIS data, Atmos. Meas. Tech., 13, 6459–6472, https://doi.org/10.5194/amt-13-6459-2020, 2020.

Mei, L., Vandenbussche, S.,Rozanov,V., Proestakis,E., Amiridis,V., Callewaert,S., Vountas, M.,Burrows,J.P., On the retrieval of aerosol optical depth over cryosphere using passive remote sensing, Remote sensing of Environment, 241, 111731, https://doi.org/10.1016/j.rse.2020.111731, 2020

2.  Atmospheric correction is another important issue that requires serious consideration. A revised version of this paper merged with the promised part II must include a discussion of this issue.

Response: Part 2 is not just promised, but already online as part 1. Part 1 and part 2 are companion papers. We are extending the validation in part 2, the new validation includes sites listed below. In the revised version of part 2, the validation is extended from one month during SnowEx17 campaign to a time period of about couple of years over different locations worldwide. And the comparison between the validation results over different regions, for instance, over Japan (can be very polluted) and Dome C (quite clean) site, can provide a deep understanding of potential impact of atmospheric correction.

[Figure]

Fig. 1 Geographic distribution of the validation sites. The colors represent the type of each site while the observation period used in this manuscript is indicated near each site.

3. The description of the SLSTR instrument in Section 2.1 lacks important information about the spectral response of the two channels at 0.55 μm and 1.6 μm. As discussed by Chen et al. (2017) the response function of the 1.6 μm channel requires careful consideration because the optical properties of snow/ice varies considerably over small wavelength intervals in the 1.6 μm spectral range.

Response: In the revised version, one more section to investigate the impact of spectral response of the two channels at 0.55 μm and 1.6 μm is included. The following figure shows the spectral response functions for 0.55 μm (left) and 1.6 μm (right). Using these spectral response functions, we performed the forward simulation with SCIATRAN model, to get TOA reflectance at 0.55 and 1.6 μm. After that, the retrieval using XBAER algorithm is performed. Since in XBAER algorithm, we did not take the spectral response functions into account, this investigation shows the impact of the spectral response function on the retrieval results.

[Figure]

Fig. 2 Spectral response function of 0.55 (left) and 1.6 (right) μm of the SLSTR instrument

4. The SLSTR instrument has a total of nine channels. The authors provide no explanation for why they have decided to use only two of those channels for this work. It would seem that using more channels should be helpful.

Response: There are couple of criteria we considered for the selection of the optimal wavelengths, for the purpose of creation of a long-term satellite snow properties dataset with good and stable accuracy.

➢ Taking the overlap channels between AATSR and SLSTR because a consistent long-term satellite snow dataset is possible only when the same algorithm can be applied on both AATSR and SLSTR instruments. In particular, the overlap channels between AATSR and SLSTR are 0.55, 0.66, 0.87, 1.6, 3.7, 10.85, and 12μm.

➢ Picking up wavelengths, for which contribution of thermal emission can be ignored, then 0.55, 0.66, 0.87, and 1.6 μm remain.

➢ Deleting the channel 0.66μm to avoid potential impact of $O_3$ absorption, after that, 0.55, 0.87, and 1.6 μm remain.

➢ Taking into account, that the retrieval algorithm is a two-stage algorithm, namely, first it uses channel with minimum impact of ice crystal shape to retrieve the grain size, and then it selects the shape using channel with minimum impact of grain size. Accounting for that the channel 0.87μm is impacted by both size and shape, 0.55 and 1.6μm channels were picked up for the retrieval.

Additionally, the AATSR/SLSTR instrument contain dual-view observation capability, therefore two wavelengths with dual-viewing at each wavelength provide enough information about our target parameters. The explanations are included in our revised version.

5.    In Section 2.1 the authors state "The statistical analysis has been performed using observations over Greenland during April and September 2017. April and September are reported to be representativeness months of the Arctic..." This focus on the months of April and September only is a serious limitation. We would like to know how the snow properties evolve over the summer season from the beginning of the melt in April/May to the freeze-up in August/September.

Response: The statistical analysis is not performed with respect to the snow properties, but for the SLSTR observation geometries (do not change with respect to meteorological conditions such as temperature). More specifically, it was performed with respect to the solar zenith angle, viewing zenith angle and relative azimuth angle. As we already discussed in the manuscript, the reason why this statistical analysis of angles was performed, and specifically for the SLSTR instrument, is to use in the sensitivity study realistic observation geometries, rather than any "arbitrary values". This enables our sensitivity study fits best and reveals most of the retrieval features based on the SLSTR instrument. And our investigations show that April and September can represent the SLSTR observation characteristics very well. The change of the underlying snow properties plays no role in this statistical analysis. So, the question with respect to melting snow in selected months is irrelevant.

6.    In Section 4 describing the XBAER algorithm, the authors state "The first stage includes the estimation of SGS using the effective Lambertian surface albedo after atmospheric correction . . .". The authors must explain what is meant by "effective Lambertian surface albedo" and also how "atmospheric correction" is performed. This information is essential.

Response: The Top Of the Atmosphere (TOA) reflectance can be described as following:

$$R_{TOA} = R_{atm} + \frac{TA}{1-SA}, \quad (1)$$

where $R_{TOA}$ is the satellite observed TOA reflectance, $R_{atm}$ , $T$ and $S$ are the atmospheric reflectance with underlying black surface, total atmospheric transmittance and spherical albedo, respectively, $A$ is the effective Lambertian surface albedo. The reason for the usage of term effective is that Eq (1) is valid under the assumption of underlying Lambertian surface. However, in reality, the surface reflection is non-Lambertian, and the parameter $A$ in Eq (1) depends on the illumination/observation geometries.

If the atmospheric parameters are known, Eq (1) can be analytically solved with respect to the parameter $A$:

$$A = \frac{(R_{TOA}-R_{atm})}{(R_{TOA}-R_{atm})S+T} \quad (2)$$

This step is usually called the atmospheric correction.

7.  The modeling carried out in this paper is based on the assumption that the snowpack is vertically homogeneous and consisting of a mono-dispersion of snow particles of a pre- defined shape. In reality these assumptions are not fulfilled. Also, the light penetration depth at wavelength 1.6 µm is much shorter than at 0.55 µm. The impact of these circumstances are not discussed.

Response: In order to assess the impacts of snowpack vertical inhomogeneity and the habit mixture on the accuracy of the retrieval algorithm, a new section is included in the revised version. The forward simulation of TOA reflectance at 0.55 and 1.6 µm is performed using the vertical profile of grain size, particle size distribution, and habit mixture as presented in the following figure. The snow grain size profile was obtained during the SnowEx17 campaign (panel (a)). The particle size distribution of the ice crystal and the habit mixture are provided by Saito et al (2019) (see panel (b) and (c)). Then the retrieval is performed assuming that the snowpack is vertically homogeneous and consisting of mono-disperse snow particles of single shape, and the retrieval accuracy is assessed.

[Figure]

Fig. 3 Snow properties used for simulations to investigate the impacts of snow profiles and mixture of different snow shapes on XBAER retrieval (a) snow grain size profile observed during SnowEx17 (b) particle size distribution of snow grain size (c) ratio of snow particle shape. (b) and (c) are suggested by Satio et al (2019)

Saito, M., P. Yang, N. G. Loeb, and S. Kato: A novel parameterization of snow albedo

based on a two-layer snow model with a mixture of grain habits, J. Atmos. Sci., 76, 1419–1436, 2019.

**Specific Comments**

• The Bidirectional Reflectance Factor (BRF) is introduced at line 206, but defined only at line 307.

Response: We shifted the full name earlier in the revised version.

• The authors state (line 445) that field-measurements of SSA are based on the assumption that snow grains have spherical shapes. What are the uncertainties in the field-measured SSA values incurred by this assumption?

Response: It is difficult to explain the uncertainty in ground-based measurements because it depends also on which technique and instrument are used for the ground-based measurements. The most important thing is that we must aware that there is some inherent difference between ground-based measurements and satellite retrievals for SSA. We have shifted the very complicated derivation of the dependence of SSA on ice crystal shape from the appendix into the main paper in the revised version, we believe, this will give the reader a better and deeper understanding.

References
Chen, N., W. Li, T. Tanikawa, M. Hori, T. Aoki, and K. Stamnes, Cloud mask over snow/ice covered areas for the GCOM-C1/SGLI cryosphere mission: Validations over Greenland, J. Geophys. Res. Atmos., 119, 12,287-12,300, 2014. doi: 10.1002/2014JD022017.

Chen, N., W. Li, T. Tanikawa, M. Hori, R. Shimada, Te. Aoki, and K. Stamnes, Fast yet accurate computation of radiances in shortwave infrared satellite remote sensing channels, Opt. Express, 17, 649-664, 2017.

Chen N., W. Li, C. Gatebe, T. Tanikawa, M. Hori, R. Shimada; T. Aoki, and K. Stamnes, New cloud mask algorithm based on machine learning methods and radiative transfer simulations, Remote Sensing of the Environment, 219, 62-71, 2018.

---

## Editor Decision (ED1)

2021-02-11

Submission tc-2020-269

Thank you for your submission to be considered for publication in The Cryosphere. The authors did significant amount of work to improve from the initial version of the paper.

I will provide here several comments that may help further polish the manuscript.

- In the revised version the authors suggest to move one appendix into the main paper in order to enhance the originality of the paper as well as complementarity with its companion paper. However, the appendix is mainly a list of equations which would, be in appendix. The authors need to clarify the rationale of what was used from the appendix;

- I believe parts of the introduction must be strengthened. For example, in lines 45-48 there is a statement on the use of satellite snow products being important for climate change study. This is rather general, which product and how are they used given that the following paragraph jumps right into SSA. The is a gap on true motivation and lingering uncertainties this paper tries to address;

- The addition of the SnowEx data on vertical profiles of snow grains lacks explanation at this stage. Grain size are reported in µm, but the SnowEx campaign included vertical SSA measurements. The sizes indicate to me a first half of the snowpack made from rounded grains, with some faceting near the bottom. Sizes displayed do not suggest depth hoar, but column are suggested (solco) in the habit fraction. Further details are needed on the linkages between 3 a and c, and the representativeness of a. Was a computed average of SnowEx profiles, or only one.

Minor comments:

- Improve quality and resolution of figures in general;
- The oscillations in Fig 2 and 3, also noted by one of the reviewers. The answer provided simply refers to the fact that is comes from the original database, but this does not answer the question. Perhaps a sentence to be added on why would avoid confusion.

Regards,

Prof. Dr. Alexandre Langlois

Associate editor, *The Cryosphere*